# Token-Level LLM Collaboration via FUSIONROUTE

**Nuoya Xiong**[1 2]   **Yuhang Zhou**[3]   **Hanqing Zeng**[3]   **Zhaorun Chen**[4]   **Furong Huang**[5]   **Shuchao Bi**[6]
**Lizhu Zhang**[3 †]   **Zhuokai Zhao**[3 †]

## Abstract

Large language models (LLMs) exhibit strengths across diverse domains. However, achieving strong performance across these domains with a *single* general-purpose model typically requires scaling to sizes that are prohibitively expensive to train and deploy. On the other hand, while smaller domain-specialized models are much more efficient, they struggle to generalize beyond their training distributions. To address this dilemma, we propose FUSIONROUTE, a robust and effective token-level multi-LLM collaboration framework in which a lightweight router simultaneously (i) selects the most suitable expert at each decoding step and (ii) contributes a complementary logit that refines or corrects the selected expert's next-token distribution via logit addition. We provide a theoretical analysis showing that pure expert-only routing is fundamentally limited unless strong global coverage assumptions hold. We solve this issue by augmenting expert selection with a trainable complementary generator, FUSIONROUTE expands the effective policy class and enables recovery of optimal value functions under mild conditions. Empirically, across both Llama-3 and Gemma-2 families and diverse benchmarks spanning mathematical reasoning, code generation, and instruction following, FUSIONROUTE outperforms both sequence- and token-level collaboration, model merging, and direct fine-tuning, while remaining competitive with domain experts on their respective tasks. The code is available at https://github.com/xiongny/FusionRoute

---

[1]Work done during the internship at Meta [2]Carnegie Mellon University [3]Meta AI [4]University at Chicago [5]University of Maryland [6]Meta TBD Lab. Correspondence to: Zhuokai Zhao <zhuokai@meta.com>, Lizhu Zhang <lizhu@meta.com>, Nuoya Xiong <nuoyax@andrew.cmu.edu>.

*Proceedings of the 43rd International Conference on Machine Learning*, Seoul, South Korea. PMLR 306, 2026. Copyright 2026 by the author(s).

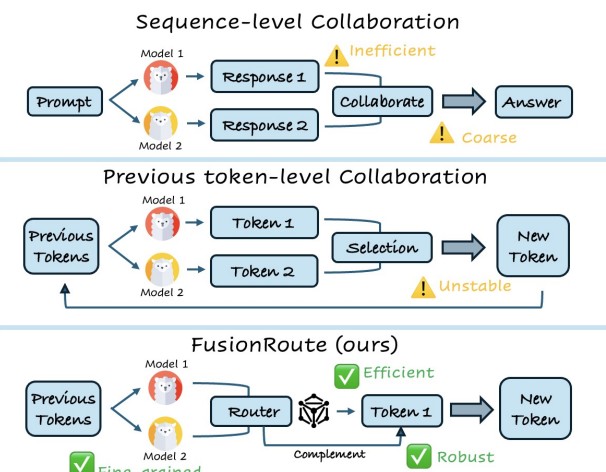

*Figure 1.* Sequence-level collaboration is coarse and inefficient, while prior token-level methods are unstable. FUSIONROUTE achieves fine-grained, efficient, and robust token-level collaboration through complementary routing.

## 1. Introduction

Large language models (LLMs) have demonstrated strong performance across an extensive range of tasks, such as mathematical reasoning (Wang et al., 2025b; Zhao et al., 2025; Zhou et al., 2025), code generation (Deng et al., 2025; Huynh & Lin, 2025; Jiang et al., 2024b; Jimenez et al., 2023), and instruction following (Wang et al., 2025a; Qin et al., 2024). Although sufficiently large, general-purpose LLMs can often deliver balanced performance across diverse domains (Hurst et al., 2024; Team et al., 2025; Yang et al., 2025; Zeng et al., 2025), their computational and monetary costs make them not ideal for every real-world applications. This motivates an alternative paradigm: employing multiple specialized LLMs, each excelling in a particular domain while remaining comparatively efficient. However, an LLM that performs exceptionally well on one task may underperform on others due to inductive biases (Levine et al., 2021; Si et al., 2023) and domain-specific training distributions (Yuan et al., 2023). As a result, achieving general-purpose performance through a collection of specialized models hinges on developing mechanisms that can robustly coordinate and leverage their complementary exper-

tise, which has emerged as a central challenge in building efficient and broadly capable LLM systems.

A natural direction toward such collaboration is mixture-of-experts (MoE), in which multiple experts are integrated into a unified architecture and trained jointly with a routing network (Zhou et al., 2022; Xue et al., 2024; Jiang et al., 2024a; Zeng et al.). While effective, this framework is expensive and inflexible since it typically needs gradient access to all experts and substantial additional end-to-end training, and it usually requires expert models to have similar structures. Another line of work aims to combine the strengths of specialized models through multi-agent systems (MAS), where different LLMs are assigned different roles with the goal of leveraging the best capabilities of each model or agent during generation (Du et al., 2023; Zheng et al., 2025). However, without prior knowledge of which types of questions each model is best suited for, determining how tasks should be allocated in advance becomes inherently challenging. Moreover, existing MAS (Liu et al., 2024; Liang et al., 2024; Ashiga et al., 2025; Huang et al., 2024; Chen et al., 2025) typically operate at a coarse response level, where each agent independently generates a full response and a final answer is selected, merged or subjected to debate post hoc. This approach is inefficient, as it requires multiple full-sequence generations, and can even degrade performance when more agents and discussions are involved due to the heavily increased context length (Zheng et al., 2025). A third direction is model merging (Yang et al., 2024; He et al., 2025), which combines multiple specialized models into a single set of parameters. Although training-free and architecturally simple, model merging is often sensitive to hyperparameters and suffers from parameter interference, making it unable to robustly emphasize different expert behaviors in changing scenarios.

To overcome these limitations, recent work has explored *token-level* multi-agent collaboration, a more fine-grained and dynamic paradigm that allows multiple LLMs to jointly produce each token during generation (Chakraborty et al., 2025; Shen et al., 2024). However, the effectiveness of these works depends heavily on the quality of the underlying models, making them not robust when expert models perform poorly, or the selection strategy is not correct on certain tasks or tokens. This lead to the following key question:

*Can we develop a token-level collaboration paradigm that works robustly, efficiently, and automatically across all scenarios?*

To address this question, we propose **FUSIONROUTE**, a token-level collaboration framework that unifies *expert selection* and *knowledge complementation* within a single, lightweight router LLM. FUSIONROUTE learns to identify which specialized model is most suitable for generation each

next token, enabling fine-grained, context-dependent use of expert capabilities without requiring expensive joint training or per-token evaluation across multiple models (Zhou et al., 2022). At the same time, FUSIONROUTE is trained to provide its own *complementary* generation signal, allowing it to refine an expert whenever the expert is uncertain or unreliable. By fusing the router's corrective logits with those of the selected expert during decoding, FUSIONROUTE achieves both robustness by mitigating expert failures, and efficiency through avoiding the overhead that limits prior multi-LLM collaboration methods. This design allows FUSIONROUTE to function as an automatic, domain-agnostic coordination mechanism, enabling consistent performance improvements across diverse tasks and datasets. In summary, we have the following contributions:

- We propose FUSIONROUTE, a token-level approach that automatically selects the most suitable expert model at each decoding step while simultaneously providing a complementary logit for better generation.
- Theoretically, we show that expert-only token-level collaboration cannot attain the optimal value function unless strong assumptions hold, revealing a fundamental limitation of prior methods. In contrast, FUSIONROUTE's complementary generator overcomes this limitation and enables recovery of the optimal policy.
- Empirically, FUSIONROUTE consistently outperforms strong baselines—including token-level collaboration methods, model merging, and finetuned models—across diverse expert domains (mathematics, coding, and instruction following) as well as general-purpose evaluations. These results highlight FUSIONROUTE's robustness, efficiency, and broad applicability in scenarios where expert strengths differ.

**Related Works** Due to space constraints, we defer a detailed discussion of related work to Appendix A, which includes many previous works for multi-LLM collaboration and multi-model integration.

## 2. Preliminaries

We formalize the decoding process of a language model as sampling from an autoregressive policy $\pi$. Let $\mathcal{X}$ denote the space of prompts and $\mathcal{Y}$ the vocabulary. A response is represented as a sequence $y = (y_1, \ldots, y_T) \in \mathcal{Y}^T$. Given a prompt $x \in \mathcal{X}$, the language model induces a conditional distribution over responses through the policy $\pi(y \mid x) = \prod_{t=1}^{T} \pi(y_t \mid x, y_{<t})$, where $\pi(y_t \mid x, y_{<t})$ specifies the probability of generating token $y_t$ given the prompt and the preceding tokens. At each step $t$, greedy decoding selects the token with the highest conditional probability under the policy $\pi$: $y_t = \arg\max_{y \in \mathcal{Y}} \pi(y \mid x, y_{<t})$.

# 3. FUSIONROUTE

## 3.1. Overall Design

FUSIONROUTE enables token-level collaboration among specialized LLMs by selecting the most suitable expert at each decoding step while remaining robust to unreliable experts. However, pure expert selection is often brittle, as even strong specialists can fail on some tokens. FUSION-ROUTE addresses this challenge by introducing a router that simultaneously selects the appropriate expert and supplies a complementary logit to refine the expert's prediction.

The FUSIONROUTE router model $\pi_\theta$ is post-trained from a base LLM parameterized by $\theta_{\mathrm{LM}}$. Given a prompt $x$ and a partial generation $y_{\leq t}$, FUSIONROUTE processes the sequence and produces two outputs: a vector of routing weights $w_\theta \in \mathbb{R}^n$, which determines the preferred expert from a set of specialized LLMs $\{\pi_1, \ldots, \pi_n\}$, and a set of logits $\log \pi_{\theta_{\mathrm{LM}}}(\cdot \mid x, y_{\leq t})$, which act as a complementary corrective component. The routing weights are generated via a lightweight linear projection applied to the final hidden state $h_{\theta_{\mathrm{LM}}}(x, y_{\leq t})$, namely $\langle W, h_{\theta_{\mathrm{LM}}}(x, y_{\leq t}) \rangle$.

During inference, FUSIONROUTE first selects the expert with the highest routing weight, $I_\theta^* = \arg\max_i w_{\theta, i}$, and uses $\pi_{\mathrm{expert}} = \pi_{I_\theta^*}$ as the selected specialist model for the current step. The final next-token distribution is then obtained by combining the router's *complementary* logits with those of the selected expert through logit addition,

$$
\begin{aligned}
\log \pi_{\mathrm{final}}&(\cdot \mid x, y_{\leq t}) \\
&= \log \pi_{\theta_{\mathrm{LM}}}(\cdot \mid x, y_{\leq t}) + \log \pi_{\mathrm{expert}}(\cdot \mid x, y_{\leq t}). \quad (1)
\end{aligned}
$$

This design preserves the domain-specific strengths of the selected expert while allowing the router to correct expert behavior when necessary.

## 3.2. Training FUSIONROUTE

Having introduced the framework of FUSIONROUTE, we now describe how the router is trained. Training FUSION-ROUTE is non-trivial since the router must *simultaneously* satisfy two coupled objectives: 1) it must reliably select the most suitable expert at each token, while 2) also providing a complementary logit contribution that corrects expert failures when necessary. Naively optimizing these objectives together can lead to bad behavior, where improvements in complementary generation degrade routing quality or vice versa. To address this challenge, we use a decoupled training strategy consisting of an SFT phase followed by a complemented preference-optimization phase.

### 3.2.1. SFT PHASE

We first perform SFT to establish two foundational properties of the router: next-token prediction capability and token-level expert selection. Importantly, this phase is not for training the router's complementary behavior, but rather to produce a stable routing mechanism aligned with expert specialization. We jointly optimize the base LLM parameters $\theta_{\mathrm{LM}}$ and the routing projection $W$ using a combination of a standard language modeling loss and a routing loss. The language modeling loss $\mathcal{L}_{\mathrm{LM}}(\theta_{\mathrm{LM}}; x, y)$ follows the standard SFT objective to encourage accurate next-token prediction on the training data.

To enable token-level routing, we introduce a routing loss that favors experts whose predictions align with the ground-truth sequence. However, many tokens (e.g., punctuation or function words) provide little routing signal because all experts predict them similarly. Supervising on such tokens can dominate the gradient and bias the router toward uninformative agreement. We therefore restrict routing supervision to token positions where experts disagree.

For a given prefix $(x, y_{\leq t})$, let $y_{i, t+1} = \arg\max_y \log \pi_i(y \mid x, y_{\leq t})$ denote the next-token prediction of expert $\pi_i$. We define the *informative* token set as $\mathcal{S} = \{t \in [T] : \exists i \neq i'$ such that $y_{i, t+1} \neq y_{i', t+1}\}$. For these positions, at token $t \in \mathcal{S}$, we first compute the router weight $w_\theta^t$ and compute the aggregated logits as $\log \pi_{\mathrm{agg}}(\cdot \mid x, y_{\leq t}) = \sum_{i=1}^n w_{\theta, i}^t \log \pi_i(\cdot \mid x, y_{\leq t})$, where the weight $w_\theta^t = (w_{\theta, 1}^t, \cdots, w_{\theta, n}^t)$ represents the output of the linear layer under parameter $\theta$ for token $t$. Then, we define the routing loss as $\mathcal{L}_{\mathrm{expert}}(\theta; x, y) = \sum_{i \in S} -\log \pi_{\mathrm{agg}}(y_i \mid x, y_{\leq i-1})$. Given a hyperparameter $\lambda$ for balancing the loss scale, the final SFT objective is

$$
\mathcal{L}_{\mathrm{SFT}}(\theta) = \mathbb{E}_{(x, y) \in \mathcal{D}_{\mathrm{SFT}}} \left[ \mathcal{L}_{\mathrm{LM}}(\theta_{\mathrm{LM}}; x, y) + \lambda \mathcal{L}_{\mathrm{expert}}(\theta; x, y) \right], \quad (2)
$$

### 3.2.2. COMPLEMENTED DIRECT PREFERENCE OPTIMIZATION (CDPO)

While SFT establishes routing and preserves next-token prediction, it does not address how the router should correct locally suboptimal expert outputs. In practice, experts may produce unreliable logits even when correctly selected. We therefore introduce a preference optimization phase that trains the router to learn complementary logit contributions while treating expert outputs as fixed. We refer to this preference-optimization stage as Complemented Direct Preference Optimization (CDPO).

During inference, the final policy is obtained by combining the router and the selected expert model logits through logits addition, as shown in Eq. (1). To align training with this decoding behavior, we introduce a *preference-optimization objective* that modifies Direct Preference Optimization (DPO) (Rafailov et al., 2023) to the router's base model parameters $\theta_{\mathrm{LM}}$. Given preference pairs $(x, y^w, y^l)$ for the same prompt $x$ and the router $\pi_{\mathrm{ref}}$ after the SFT

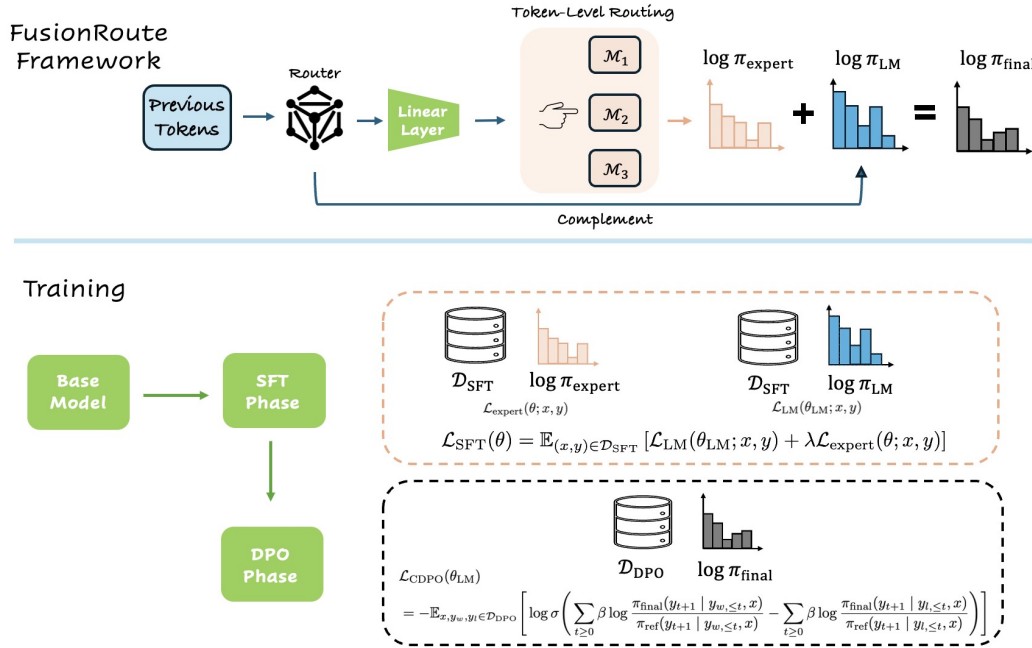

*Figure 2. Top:* Overall architectural design of FUSIONROUTE. FUSIONROUTE enables multiple specialized LLMs to work collaboratively through a carefully designed router. The router outputs *both* the weights for token-level routing and the logits to complement the experts' output. *Bottom:* The training process is decomposed into two phases. In the SFT phase (§3.2.1), the router learns token-wise mixture weights over the expert models and is jointly fine-tuned to become a good response generator. In the CDPO phase (§3.2.2), FUSIONROUTE refines the final policy by using preference optimization to improve upon the experts' outputs.

phase, for any policy $\pi$, define the contrastive loss as

$$r_\pi(x, y, y_l) := \sum_{t \geq 0} \beta \log \frac{\pi(y_{t+1} \mid y_{w, \leq t}, x)}{\pi_{\text{ref}}(y_{t+1} \mid y_{w, \leq t}, x)}$$
$$- \sum_{t \geq 0} \beta \log \frac{\pi(y_{t+1} \mid y_{l, \leq t}, x)}{\pi_{\text{ref}}(y_{t+1} \mid y_{l, \leq t}, x)}.$$

The objective is defined by

$$\mathcal{L}_{\text{CDPO}}(\theta_{\text{LM}})$$
$$= -\mathbb{E}_{(x, y_w, y_l) \sim \mathcal{D}_{\text{DPO}}} \left[ \log \sigma \left( \ell_{\pi_{\text{final}}}(x, y_w, y_l) \right) \right]$$
$$= -\mathbb{E}_{(x, y_w, y_l) \sim \mathcal{D}_{\text{DPO}}} \left[ \log \sigma \left( \ell_{\pi_{\theta_{\text{LM}}}}(x, y_w, y_l) \right. \right.$$
$$\left. \left. + \underbrace{\sum_{t \geq 0} \beta \log \frac{\pi_{\text{expert}}(y_{t+1} \mid y_{w, \leq t}, x)}{\pi_{\text{expert}}(y_{t+1} \mid y_{l, \leq t}, x)}}_{\text{(Bias)}} \right) \right], \quad (3)$$

where $\sigma(x) = (1 + e^{-x})^{-1}$ is the sigmoid function, and $\beta$ controls the strength of preference alignment.

Importantly, we do not backward propagate any gradient for the bias term. It can be regarded as an extra token-level bias. Optimizing this loss naturally encourages the router to provide a complementary logit contribution that compensates for the weaknesses in the expert models. When

an expert is strong for a prompt, its log-ratio gap term (B) is large, yielding a small gradient on $\theta_{\text{LM}}$. When the expert is weak, the gap term is small, leading to a larger gradient that encourages stronger corrective logits. As a result, the router refines expert predictions precisely when needed, improving robustness while aligning training with the collaborative decoding behavior used at inference time.

### 3.2.3. MIX TRAINING

Since preference optimization updates the router's base parameters $\theta_{\text{LM}}$, it can indirectly affect expert selection through shared representations. It is therefore necessary to preserve both reliable expert selection and effective complementary logit refinement during training. However, naively applying DPO to the entire router often degrades routing, as the final linear layer overfits to preference signals and loses its ability to select experts correctly.

To address this issue, we adopt a decoupled optimization strategy. Specifically, for SFT samples, all parameters, including the routing layer, are updated using the routing loss $\mathcal{L}_{\text{LM}}$. On the other hand, we apply the preference-optimization objective (Eq. (3)) only to the router's base model parameters $\theta_{\text{LM}}$, while excluding the routing projection. To ensure consistency between the base model and the linear layer, we jointly mix preference-optimization data with supervised SFT data during training. This mixed

---

**Algorithm 1** Mix Training

---

1: **Initial**: Dataset $\mathcal{D}_{\text{SFT}}$, $\mathcal{D}_{\text{DPO}}$, SFT router $\pi_\theta$, batch size $B$, learning rate $\eta$.
2: Mix two datasets randomly to get $\mathcal{D}_{\text{mix}}$.
3: **for** $t = 1, 2, \cdots ,$ **do**
4:     Receive a batch of data points $\{d_1, \cdots , d_B\} \in \mathcal{D}_{\text{mix}}$. Set $\mathcal{L} = 0$.
5:     **for** $i = 1, 2, \cdots , B$ **do**
6:        **if** $d_i = (x, y) \in \mathcal{D}_{\text{SFT}}$ **then**
7:           $\mathcal{L} = \mathcal{L} + \lambda \mathcal{L}_{\text{LM}}(\theta; d_i)$.
8:        **else**
9:           $\mathcal{L} = \mathcal{L} + \mathcal{L}_{\text{CDPO}}(\theta_{\text{LM}}; d_i)$.
10:        **end if**
11:     **end for**
12:     Update the parameter $\theta = \theta - \eta \nabla_\theta \mathcal{L}$.
13: **end for**

---

training scheme preserves expert-selection capability while enabling the router to acquire a complementary logit contribution. The full procedure is summarized in Algorithm 1.

## 4. Theoretical Analysis

### 4.1. Token-Level Markov Decision Process

We formulate the decoding process as a token-level Markov Decision Process (MDP) $\mathcal{M} = \{\mathcal{S}, \mathcal{A}, \mathbb{P}, r\}$, where $\mathcal{S}$ is the state space, $\mathcal{A}$ is the action space, $\mathbb{P}$ is the transition kernel, and $r$ defines the reward function. Let the prompt space be $\mathcal{X}$ and the vocabulary be $\mathcal{Y}$. Starting from the prompt $s_0 = x$, at step $t$ the state is defined by $s_t = (x, y_{\leq t})$, and the action is represented by the next token $a_t = y_{t+1}$. The transition kernel is then defined by $\mathbb{P}(s_{t+1} \mid s_t, a_t) = \mathbb{I}\{s_{t+1} = (x, y_{\leq t}, y_{t+1})\}$. Finally, the token-level reward function $r(s, a) = r(x, y_{\leq t+1}) \in [0, 1]$ maps any text $(x, y_{\leq t+1})$ to a real number. A language model is a policy $\pi(a \mid s) = \pi(y_{t+1} \mid x, y_{\leq t})$, where it takes the previous text $(x, y_{\leq t})$ as the input and generates the next token $y_{t+1}$. Let $\tau = (x, y_1, \cdots , y_T)$ denote the trajectory, which is indeed a full response generated by a language model. The value function $V^\pi(s) = V^\pi(x, y_{\leq t})$ for a state $s = (x, y_{\leq t})$ can be defined as $V^\pi(x, y_{\leq t}) = \mathbb{E}_{\tau \sim \pi(\cdot|x, y_{\leq t})} \left[ \sum_{i=t+1}^T r(x, y_{\leq i}) \right]$. Similarly, for $a = y_{t+1}$, the Q function can be defined as $Q^\pi(x, y_{\leq t}, y_{t+1}) = r(x, y_{\leq t+1}) + \mathbb{E}_{\tau \sim \pi(\cdot|x, y_{\leq t+1})} \left[ \sum_{i=t+1}^T r(x, y_{\leq i+1}) \right]$. We assume a upper bound $Q_{\max}$ and $V_{\max}$ for any Q-function and value function respectively.

### 4.2. Token-Level Expert Selection through the Lens of Performance Difference Lemma

In this section, we establish a connection between the Performance Difference Lemma (PDL) (Kakade & Langford,

2002) and our token-level expert selection training. At a high level, PDL characterizes how deviations from the optimal policy at individual decision steps accumulate into a final performance gap.

**Lemma 4.1** (Performance Difference Lemma (Kakade & Langford, 2002)). *Denote $\rho \in \Delta(\mathcal{X})$ as a distribution over the prompt space. Suppose each response $y$ has length $T$. Then, for any two policies $\pi, \pi^*$, we have*

$$\mathbb{E}_{x \sim \rho} \left[ V^{\pi^*}(x) - V^\pi(x) \right] = \sum_{t=0}^{T-1} \mathbb{E}_{x \sim \rho, y_{\leq t} \sim \pi} \left[ V^*(x, y_t) - \mathbb{E}_{y_{t+1} \sim \pi(\cdot|x, y_{\leq t})}[Q^*(x, y_{\leq t+1})] \right].$$

When applied to LLM, this lemma shows that selecting an expert whose next token approximately maximizes the optimal value at each token yields a near-optimal final policy.

To formalize the connection, we assume the SFT dataset is generated by the optimal policy $\pi^*(\cdot \mid s)$, where $\pi^*$ achieves the optimal value function in the token-level MDP, and we denote $Q^{\pi^*}$ as the optimal Q-function. Since we trained on the SFT dataset which follows trajectories generated by $\pi^*$, at step $t$ with prefix $(x, y_{\leq t})$, the router can be viewed as approximating an expert action that aligns with the action preferred by $\pi^*$, i.e., choosing the expert that can maximize the optimal Q-function: $\pi_{\text{expert}}(\cdot \mid x, y_{\leq t}) = \pi_I(\cdot \mid x, y_{\leq t})$, where $I = \arg \max_i \mathbb{E}_{y_{t+1} \sim \pi_i(\cdot|x, y_{\leq t})}[Q^{\pi^*}(x, y_{\leq t+1})]$.

We make the following assumption to state that for any prefix $x, y_{\leq t}$, there is one correct model that achieves approximate optimal value function.

**Assumption 4.2** (Global Coverage Assumption). For any $x, y_{\leq t}$, there exists a small $\Delta \leq 0$ and an $i \in [n]$ such that

$$\left| \mathbb{E}_{y_{t+1} \sim \pi_i(\cdot|x, y_{\leq t})}[Q^{\pi^*}(x, y_{\leq t+1})] - V^*(x, y_{\leq t}) \right| \leq \Delta, \quad (4)$$

This assumption states that the expert set is assumed to be sufficiently expressive so that for each token position, one expert always achieves an approximately optimal value. Following the Assumption 4.2 and Lemma 4.1, we can get

$$\mathbb{E}_{x \sim \rho} \left[ V^{\pi^*}(x) - V^\pi(x) \right] \leq \sum_{t=0}^{T-1} \Delta = T\Delta, \quad (5)$$

which means that when $\Delta = o(1/T)$, an idealized token-level routing policy can achieve near-optimal expected return under by selecting the appropriate expert at each step. We also discuss the difference between our approach and the prior token-level collaboration approach Collab (Chakraborty et al., 2025) in Appendix C.

However, Assumption 4.2 is relatively strong, as it requires global coverage over *all* prefixes, including those unseen in supervised training. A more natural alternative is to assume coverage only on the trained data generated by $\pi^*$. However,

we will show that this weaker assumption is insufficient to guarantee approximately optimal responses, revealing a fundamental limitation of purely token-level collaboration.

## 4.3. Limitations of Purely Token-Level Collaboration

In the previous subsection, we showed that router training can recover the ideal action, with a relative restrictive Assumption 4.2. A more reasonable requirement is to enforce Eq. (4) only on *good* prefixes $(x, y_{\leq t})$. We consider two types of good prefixes: (i) prefixes sampled from the optimal policy $\pi^*$, corresponding to the *single-policy coverage* assumption; and (ii) prefixes that can potentially lead to high-quality responses, corresponding to the *generalization coverage* assumption. Together, these prefixes capture the text patterns commonly encountered during training. Under these assumptions, we show that although a near-optimal candidate selection may exist, it cannot be recovered from observations of the value function $Q^{\pi^*}$ alone.

**Theorem 4.3** (Informal). *Suppose the observation space only contains the prefix $(x, y_{\leq t})$ and the optimal Q-value function for previous text $\{Q^*(x, y_{\leq k})\}_{k=1}^{t}$ and the all possible next token $\{Q^{\pi^*}(x, y_{\leq t}, y)\}_{y \in \mathcal{Y}}$. Then, even if (i) there exists a near-optimal trajectory achievable via purely token-level routing, and (ii) the value gap is bounded by a small $\Delta$ for all good prefixes, no token-level routing algorithm can guarantee a near-optimal policy.*

The formal statement and the proof is provided in Appendix B.1. The impossibility in Theorem 4.3 stems from an identifiability failure, where observing optimal values along trajectories generated by $\pi^*$ is insufficient to determine which expert actions actually realize those values. In fact, Theorem 4.3 shows that using SFT to train a router that chooses the optimal model is not reliable, because this is essentially equivalent to use behavior cloning for learning the actions that maximize the optimal value function $Q^{\pi^*}$.

## 4.4. Benefits of Router Training

The previous theorem shows that, even if a prefix has high value under the optimal policy $\pi^*$, since each token must be chosen from a fixed set of expert actions, any misalignment between $\pi^*$ and the experts at some tokens leads to uncontrollable approximation error. To address this issue, we add the logits of the router as a complementary component (Eq. (1)) to the final logits and train the final logits to align with the optimal policy. This removes the need to assume that any expert alone attains near-optimal Q-values; instead, we assume that the total variation distance between the final policy and the optimal policy is bounded. In fact, since our training is to align $\pi_i'$ with the expert data where $\log \pi_i' = \log \pi_i + \log \pi_{\text{router}}$, we can assume

$$\mathbb{E}_{x \sim \rho, y_{\leq t} \sim \pi^*}[\arg \min_i \text{TV}(\pi_i'(\cdot \mid x, y_{\leq t}), \pi^*(\cdot \mid x, y_{\leq t}))] \leq \Delta.$$

Then, if at each position $t$, the policy $\pi$ chooses the expert $i$ that $\pi_i'$ minimizes the TV distance to the optimal policy,

from PDL we can get

$$\mathbb{E}_{x \sim \rho}[V^{\pi^*}(x) - V^{\pi}(x)] \leq TQ_{\max}\Delta. \qquad (6)$$

The proof is provided in Appendix B.2. This result highlights a key conceptual shift: rather than requiring any fixed expert to be near-optimal, we allow a router to jointly select and complement experts by adding adaptive logits. The resulting policy expands the expressivity of the action space, enabling approximation of the optimal policy $\pi^*$ even when no single expert suffices.

Furthermore, this framework also has advantages compared to the directly fine-tuned model. When the optimal policy and a candidate model are already reasonably well aligned, the router's complementary component need not be highly accurate. In this case, the router is substantially easier to learn, as it only needs to correct the limited set of states where the candidate model deviates from $\pi^*$. Consequently, combining a candidate model with a trainable router can outperform both fine-tuning a single model and pure routing.

## 5. Experiments

In this section, we first describe the experimental setup in §5.1. We then present mixed-domain performance and win rates on general benchmarks in §5.2. These results show that FUSIONROUTE enables effective multi-model collaboration by selecting domain-appropriate experts while maintaining strong performance on general tasks.

### 5.1. Experimental Setup

**Baselines** We compare FUSIONROUTE against a broad set of baselines, including sequence-level collaboration, token-level collaboration, model merging, and single-model fine-tuning. For sequence-level collaboration, we compare with Sequence Selection, where each expert independently generates a full response using greedy decoding, and an external reward model selects the highest-scoring output. For the token-level collaboration, we include Collab (Chakraborty et al., 2025), which scores candidate tokens from multiple models with an external reward signal. We also include two popular model merging approaches, DARE (Yu et al., 2024) and TaskArithmetic (Ilharco et al., 2022), as well as a fine-tuned model baseline that applies the same SFT and DPO as FUSIONROUTE but removes routing and collaboration, isolating the contribution of complementary logits.

**Models and Datasets** We conduct experiments on both the Llama-3 (Dubey et al., 2024) and Gemma-2 (Team et al., 2024) families. For each family we choose three models that are expertise in math, coding and instruction-following from MergeBench (He et al., 2025) respectively. Specifically, for the Llama-3 family, we use

| Method | GSM8K | MATH500 | MBPP | HumanEval | IfEval | Avg Acc. |
|---|---|---|---|---|---|---|
| *Llama-3 Family* | | | | | | |
| *Expert Models* | | | | | | |
| Math Expert | 0.86 | 0.36 | 0.23 | 0.37 | 0.31 | 0.426 |
| Code Expert | 0.52 | 0.15 | 0.34 | 0.66 | 0.48 | 0.430 |
| Instruct Expert | 0.74 | 0.27 | 0.36 | 0.52 | 0.67 | 0.512 |
| *Collaboration Approaches* | | | | | | |
| Sequence Selection | 0.76 | 0.31 | 0.21 | 0.45 | 0.50 | 0.466 |
| Collab | 0.82 | 0.32 | 0.28 | 0.54 | 0.55 | 0.502 |
| Fine-tuned Model | 0.75 | 0.26 | 0.36 | 0.58 | **0.72** | 0.536 |
| DARE | 0.75 | 0.28 | 0.23 | 0.26 | 0.32 | 0.368 |
| TaskArithmetic | 0.82 | 0.32 | 0.24 | 0.39 | 0.35 | 0.424 |
| FUSIONROUTE (ours) | **0.82** | **0.33** | **0.36** | **0.63** | 0.69 | **0.566** |
| *Gemma-2 Family* | | | | | | |
| *Expert Models* | | | | | | |
| Math Expert | 0.67 | 0.27 | 0.22 | 0.36 | 0.47 | 0.398 |
| Code Expert | 0.39 | 0.16 | 0.27 | 0.41 | 0.52 | 0.350 |
| Instruction Expert | 0.19 | 0.08 | 0.23 | 0.34 | 0.61 | 0.290 |
| *Collaboration Approaches* | | | | | | |
| Sequence Selection | 0.62 | **0.24** | **0.27** | 0.37 | 0.54 | 0.408 |
| Collab | 0.52 | 0.22 | 0.26 | 0.35 | 0.45 | 0.360 |
| Fine-tuned Model | 0.54 | 0.17 | 0.24 | 0.34 | **0.68** | 0.394 |
| TaskArithmetic | 0.43 | 0.15 | 0.14 | 0.28 | 0.34 | 0.268 |
| DARE | 0.36 | 0.10 | 0.14 | 0.16 | 0.36 | 0.224 |
| FUSIONROUTE (ours) | **0.65** | 0.22 | 0.25 | **0.40** | 0.61 | **0.426** |

*Table 1.* Cross-domain performance comparison on Llama-3 and Gemma-2 families. Best and second-best performance within each family are highlighted by **bold** and underline, respectively.

math expert models Llama-3.1-8B-Instruct_math and coding expert models Llama-3.1-8B-Instruct_coding from MergeBench (He et al., 2025) as the two expert models, and Llama-3.1-8B-Instruct as the instruction-following expert. Following Chakraborty et al. (2025), we use the one open-sourced Ray2333/reward-model-Mistral-7B-instruct-Unified-Feedback as the external reward for Collab (Chakraborty et al., 2025). For the Gemma-2 family, we choose Gemma-2-2B-Instruct_math, Gemma-2-2B-Instruct_coding and Gemma-2-2B-Instruct_instruction from MergeBench (He et al., 2025) as three expert models. For a fair comparison, we use weqweasdas/RM-Gemma-2B as the external reward for Collab (Chakraborty et al., 2025), instead of a larger 7B reward model. We randomly select 500 samples from the PerfectBlend dataset (Xu et al., 2024) as the test set and use the remaining samples as the training pool. For the initial SFT phase, we randomly sample 200k examples from the PerfectBlend training split, which provides a balanced mixture of mathematics, coding, and instruction-following tasks. For both the Llama and Gemma families, the router is initialized from the instruction-following expert and fine-tuned by minimizing the loss in Eq. (2) with $\lambda = 1/3$ and learning rate $10^{-5}$ for one epoch. After the SFT phase, we further train the model using the mixed training procedure in Algorithm 1. We sample 100k examples from the PerfectBlend training split as $\mathcal{D}_{\text{SFT}}$ and 100k preference pairs from the OpenHer-

mes dataset as $\mathcal{D}_{\text{DPO}}$ with learning rate $10^{-5}$, $\beta = 0.1$, and $\lambda = 1/3$ for one epoch. For the directly fine-tuned baseline, we apply the same SFT and DPO procedures and hyperparameters. For model merging baselines, we use the default parameters $p = 0.9$ and $\lambda = 1$ for DARE and $\lambda = 1$ for TaskArithmetic.

### 5.2. Main Results

#### 5.2.1. MIXED-DOMAIN PERFORMANCE.

In practical deployment, users interact with a single model without knowing the prompt's domain. We therefore evaluate FUSIONROUTE in a general-purpose, mixed-domain setting, where the model must automatically leverage the most appropriate expert at inference time. We consider five benchmark datasets covering diverse expert domains: GSM8K (Cobbe et al., 2021) and MATH500 (Lightman et al., 2023) for mathematical reasoning, MBPP (Austin et al., 2021) and HumanEval (Chen et al., 2021) for code generation, and IfEval (Zhou et al., 2023) for instruction following. More implementation details for each task are provided in Appendix E.1.

As shown in Table 1, FUSIONROUTE consistently outperforms sequence-level collaboration, prior token-level collaboration Collab, model merging methods, and directly fine-tuned models. These results demonstrate that FUSION-

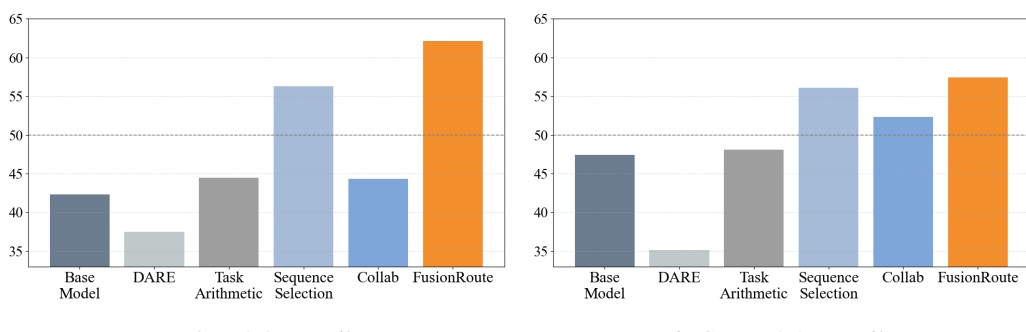

*(a)* Llama3-8B Family          *(b)* Gemma2-2B Family

*Figure 3.* GPT-4o winrate on general datasets compared to fine-tuned model.

ROUTE effectively functions as a general-purpose model assembled from specialized experts, relieving users from the need to select domain-specific checkpoints while delivering strong and stable performance across heterogeneous tasks. The examples of routing behavior in several domain-specific tasks are provided in Appendix E.2.

Importantly, FUSIONROUTE does not sacrifice specialization. In fact, FUSIONROUTE matches or exceeds domain experts on their strongest benchmarks, such as math experts on GSM8K and MATH500 or coding experts on MBPP and HumanEval, while remaining robust across mixed domains. This shows that FUSIONROUTE is compatible with expert specialization, while being substantially more robust and generalized in mixed-domain usage.

### 5.2.2. PERFORMANCE ON GENERAL DATASET

To further assess overall response quality beyond task-specific correctness, we evaluate FUSIONROUTE on a general dataset by sampling 500 prompts from the Perfect-Blend (Xu et al., 2024) test set and generating responses. We use GPT-4o to evaluate the overall response quality compared to the response of the directly fine-tuned model (Fine-tuned Model in Table 1). We compute the fraction of wins as winrate over 500 examples. More details are provided in Appendix E.1. As shown in Fig. 3, FUSIONROUTE achieves a substantially higher win rate than the fine-tuned baseline on both model families, compared to all other baselines, which indicates better overall response quality on general prompts. This improvement reflects better alignment, fluency, and formatting, complementing the strong task-level performance observed on domain benchmarks.

### 5.2.3. FUSIONROUTE IS MORE BENEFICIAL AT A LARGER SCALE

As shown in Fig. 3, the performance gap between FUSION-ROUTE and other baselines becomes markedly larger in the 8B Llama-3 family. In this setting, both Collab and Sequence Selection exhibit substantial degradation in win rate. This suggests that as model capacity grows, selecting among

fixed expert outputs becomes increasingly brittle, whereas FUSIONROUTE 's complementary routing effectively exploits the added capacity to refine expert predictions.

In contrast, in the 2B Gemma-2 family, the gap between FU-SIONROUTE and pure token-level collaboration is smaller. While other collaboration approaches still underperform FUSIONROUTE, they outperform the directly fine-tuned baseline. This indicates that there is relatively less room for complementary correction at smaller scales, due to the limited capacity of the router. Overall, these results suggest that FUSIONROUTE becomes better as the model scale grows.

## 6. Ablation Study

### 6.1. Ablation on Complementary Logit Contribution

In this section, we examine the role of the router's complementary logit contribution in FUSIONROUTE. In particular, we aim to isolate whether token-level expert selection alone is sufficient for effective collaboration. We therefore remove the complementary component, using the router after the SFT phase solely for token-level expert selection without adding router logits.

Table 2 in Appendix D reports the mixed-domain performance of the routing-only variant compared with the full FUSIONROUTE framework. We see that FUSIONROUTE consistently outperforms FUSIONROUTE w/o complementary logits across nearly all benchmarks and both model families. The performance gap is particularly pronounced on coding and instruction-following tasks, where even correctly selected experts can produce locally suboptimal or misaligned tokens that require correction. These results provide direct empirical evidence support for our theoretical analysis in §4: Even with accurate expert selection, relying solely on fixed expert logits cannot reliably yield a near-optimal policy. Allowing the router to add complementary logits expands the policy's expressivity, enabling it to correct expert failures and achieve robust performance.

Also, we find that the routing-only variant (FUSIONROUTE

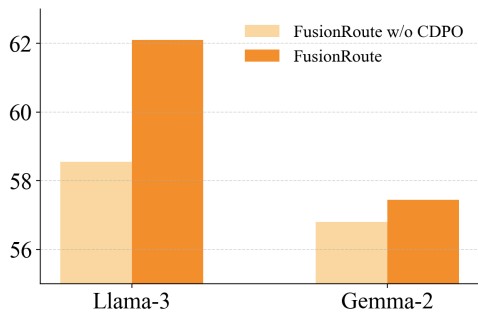

*Figure 4.* GPT-4o winrate comparison: FUSIONROUTE with vs. without CDPO on Llama-3-8B and Gemma-2-2B Family.

w/o complementary logits) already outperforms the Collab across most benchmarks. This improvement suggests that directly training the routing component on expert data is crucial for learning accurate and stable token-level routing behaviors. In contrast, controlled-decoding approaches like Collab that rely solely on external reward signals tend to exhibit higher instability.

### 6.2. Ablation on FUSIONROUTE Training Procedure

Beyond the role of complementary logits at inference time, we also analyze FUSIONROUTE 's training pipeline by ablating the preference-optimization stage to assess the effectiveness of CDPO relative to SFT. To this end, we compare two variants: a FUSIONROUTE after the SFT phase, and the full FUSIONROUTE framework with CDPO training. We evaluate these variants using GPT-4o win rate compared to the fine-tuned model baseline on the general dataset.

The result are shown in Fig. 4. We observe that while SFT already provides a reasonable initialization, applying CDPO training leads to a substantial improvement in win rate. In particular, the full FUSIONROUTE model significantly outperforms its SFT-only counterpart, indicating that the complementary component learned during CDPO effectively corrects expert failures and enhances the overall response quality. These findings suggest that the performance gains of FUSIONROUTE depend critically on the preference-optimization stage in the training procedure, which substantially improves general response quality.

## 7. Conclusion

In this paper, we present FUSIONROUTE, a framework that enables multiple candidate models to collaborate and works well automatically across diverse scenarios because of a routing mechanism, and provides robustness by letting the router provide complementary logits to refine the overall response quality. In addition, FUSIONROUTE is computationally efficient, as it avoids requiring each model to generate

candidate tokens or full responses for selection. Theoretically, we characterize the limitations of purely token-level collaboration and demonstrate the benefits of incorporating complementary logits. Extensive experiments on both domain-specific and general datasets verify the effectiveness of our framework.

## Impact Statement

This paper presents work whose goal is to advance the field of multi-LLM collaboration. There are many potential societal consequences of our work, none which we feel must be specifically highlighted here.

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

# A. Related Works

## A.1. Multi-LLM Collaboration

Recent advances in LLMs have shown that sufficiently large, general-purpose LLMs can achieve strong performance across a wide range of tasks, including mathematics, coding, and instruction following (Hurst et al., 2024; Team et al., 2025; Yang et al., 2025; Zeng et al., 2025). However, such models are often prohibitively expensive to train and deploy, motivating interest in more efficient alternatives. At the same time, smaller and domain-specialized LLMs can perform competitively within their respective domains, but typically struggle to generalize beyond the distributions on which they are trained. Bridging this gap between specialization and generalization without incurring the cost of training ever-larger models has therefore become an important challenge. One line of works study the benefits of multi-agent debate, in which several agents discuss and debate to generate a final response (Liu et al., 2024; Chan et al., 2023; Liang et al., 2024). Existing works also try to model the social and economic behavior in the real world using different specialized LLMs (Zhao et al., 2023). Moreover, model ensemble methods (Ashiga et al., 2025; Huang et al., 2024; Chen et al., 2025) combine responses or probability distributions from multiple models in a fixed manner to leverage their complementary strengths without modifying individual parameters; however, they lack dynamic, context-dependent collaboration among candidate models. More fine-grained token-level collaboration has been studied in recent years. Collab (Chakraborty et al., 2025) uses a controlled decoding-based approach to evaluate the candidate tokens during decoding. Compared to them, our decoding algorithm is much more efficient and also achieves a better performance. CO-LLM (Shen et al., 2024) also explores token-level collaboration, but mainly uses one strong math expert to assist a small fine-tuned model, and its output depends entirely on the candidate experts. In contrast, we use multiple specialized LLMs, evaluate both cross-domain performance and general win rate, and additionally train the router as a complementary generator to improve robustness.

## A.2. Multi-LLM Integration

Beyond multi-agent LLM collaboration, a substantial body of works also consider how to utilize and integrate different models to be comprehensive. The most representative area is Mixture-of-Expert (MoE), in which multiple expert models are combined for each layer with some routing network, allowing the system to leverage the specialized knowledge of each expert for different inputs (Zhou et al., 2022; Xue et al., 2024; Jiang et al., 2024a). However, because the final architecture tightly integrates the layers of multiple models with a routing network, these approaches are typically applicable only to models with similar structures. They also require access to the experts' gradients and joint training of the entire system, which limits flexibility and incurs significant additional training cost. Model merging is another approach for integrating multiple experts (Yu et al., 2024; Ilharco et al., 2022; Yadav et al., 2023). While it does not incur additional training costs, its performance is often limited and sensitive to hyperparameters. Moreover, its flexibility is also limited, as it requires all models to share the same architecture. There is also a line of work on pluralistic alignment, in which multiple models with potentially conflicting objectives, such as safety and helpfulness, are integrated to achieve tradeoffs across multiple objectives (Shi et al., 2024; Xiong & Singh, 2025; Wang et al., 2025c). In contrast to these approaches, FUSIONROUTE does not merge or jointly train expert models, nor does it require architectural compatibility or gradient access. Instead, FUSIONROUTE enables flexible, token-level integration of heterogeneous, off-the-shelf LLMs by learning a lightweight router that selects among experts and provides complementary logits during decoding. This design allows FUSIONROUTE to combine the strengths of specialized models while retaining robustness and generalization, without the rigidity or training overhead inherent in MoE and model-merging approaches.

# B. Theoretical Analysis

## B.1. Statement and Proof of Theorem 4.3

Here we provide the formal statement of Theorem 4.3 and the detailed proof.

**Theorem B.1.** *Denote $\{n_1, n_2, \cdots, n_t\}$ as a path of length $t$, where $n_i \in [n], 1 \leq i \leq t$. Assume that all $\pi^*$ and $\{\pi_i\}_{i \in [m]}$ are deterministic policy, the response length is fixed as $T$, and the transition function is also deterministic. Suppose the learner's observation space $\mathcal{O}_t$ contains*

$$o_t = \left\{ x, y_{\leq t}, \{Q^{\pi^*}(x, y_{\leq k})\}_{k=1}^t, \{Q^{\pi^*}(x, y_{\leq t}, y)\}_{y \in \mathcal{Y}} \right\}.$$

*at the visited states $(x, y_{\leq t})$ during decoding. Suppose $\Delta > 0$ is a error constant. Then, there is a MDP such that no token-level routing algorithm $\mathcal{A} : \mathcal{O} \mapsto [n]$ can achieve $V^{\pi_{\mathcal{A}}} \geq V^{\pi^*} - T/2 + 2$ for all possible rewards in this MDP, even*

*if we have the following two good properties:*

1. **Existence of a Near-Optimal Path.** *There exists a small $\varepsilon \leq \Delta \in [0, 1]$ and at least one possible token-level routing strategy $P^* = \{n_1^*, n_2^*, \cdots, n_T^*\}$ such that*

$$V^{\pi^*} = V^{P^*} + \varepsilon. \tag{7}$$

2. **Single Policy Coverage.** *For the optimal policy $\pi^*$, the error constant $\Delta \in [0, 1]$ such that the following approximation holds:*

$$\mathbb{E}_{x, y_t \sim \pi^*} \left| \arg\max_i \mathbb{E}_{y_{t+1} \sim \pi_i(\cdot | x, y_{\leq t})} \left[ Q^{\pi^*}(x, y_{\leq t+1}) \right] - \mathbb{E}_{y_{t+1} \sim \pi^*(\cdot | x, y_{\leq t})} \left[ Q^{\pi^*}(x, y_{\leq t+1}) \right] \right| \leq \Delta. \tag{8}$$

3. **Generalization Coverage** *For any prefix $x, y_{\leq t}$, if there exists a full response $x, y_{\leq T}$ with prefix $x, y_{\leq t}$ and $R(x, y_{\leq T}) \geq V^{\pi^*} - \Delta$, we have*

$$\left| \arg\max_i \mathbb{E}_{y_{t+1} \sim \pi_i(\cdot | x, y_{\leq t})} \left[ Q^{\pi^*}(x, y_{\leq t+1}) \right] - \mathbb{E}_{y_{t+1} \sim \pi^*(\cdot | x, y_{\leq t})} \left[ Q^{\pi^*}(x, y_{\leq t+1}) \right] \right| \leq \Delta. \tag{9}$$

*Proof.* We assume that the prompt $x$ is fixed and that the response length is fixed to $T$. For any $t \leq T$, define

$$\mathcal{P}_t = \left\{ (n_1, \ldots, n_t) \mid n_k \in [n], \forall k = 1, \ldots, t \right\}$$

as the set of all possible expert-selection sequences of length $t$. We further assume that all expert models and transition dynamics are deterministic. In particular, for any expert $\pi_i$ and any prefix $(x, y_{\leq t})$, the decoding rule $\pi_i(x, y_{\leq t}) \in \mathcal{Y}$ deterministically outputs the next token. Given any selection sequence $p = (n_1, \ldots, n_t) \in \mathcal{P}_t$, the induced token sequence $\mathbf{y}^p = (y_1^p, \ldots, y_t^p)$ is defined recursively as

$$y_k^p = \pi_{n_k}(x, y_{\leq k-1}^p), \qquad k = 1, \ldots, t,$$

with the convention that $y_{\leq 0}^p := \emptyset$. Accordingly, the induced state can be written as $s^p = (x, \mathbf{y}^p)$.

**Construction** Now we construct $|P_{T/2}|$ MDPs $\{M_{p_{T/2}}\}_{p_{T/2} \in P_{T/2}}$ as follows:

**Same Structure** We first define the shared components of all candidate MDPs.

The initial state is $s_0 = x$. The action space corresponds to token generation, with vocabulary $\mathcal{Y}$. Unless otherwise specified, the reward function is defined to be 1 for all prefixes $(x, y)$.

For each expert model $i \in [n]$, define

$$y_{1,i} := \arg\max_{y \in \mathcal{Y}} \pi_i(y \mid x).$$

We assign the first-step rewards as follows. For each token $y_{1,i}$, define

$$r(x, y_{1,i}) = 1 - \varepsilon. \tag{10}$$

Otherwise, for any $y \in \mathcal{Y} \setminus \{y_{1,i}\}_{i \in [n]}$, we assign $r(x, y) = 1$.

**Difference** Now we show the difference between multiple candidate MDPs $\{M_{p_{T/2}}\}_{p_{T/2} \in P_{T/2}}$. Each MDP $M_p$ is constructed in the following ways:

- For path $p' \in \{p \in P_{T/2+i} \mid i \geq 1, p \text{ has prefix } p_{T/2}\}$, we define $r(x, \mathbf{y}^{p'}) = 1$.

- For any path $p' \in \{p \in P_{T/2+1} \mid p \text{ does not have prefix } p_{T/2}\}$, $r(x, \mathbf{y}^{p'}) = 1 - \Delta$.

- For any path $p' \in \{p \in P_{T/2+i} \mid i \geq 2, p \text{ does not have prefix } p_{T/2}\}$, we define $r(x, \mathbf{y}^{p'}) = 0$.

For all other states that are not mentioned, their reward functions are defaulted to 1.

**Value Function**    Under this construction, for any prefix $(x, y_{\leq t})$, the optimal policy will choose $y_{\text{PR}}$ as its next token, and the final optimal value function satisfies

$$V^*(x, y_{\leq t}) = T - t, \quad \forall \text{ prefix } (x, y_{\leq t}).$$

This is because the optimal policy $\pi^*$ selects the reward-maximizing token at every remaining time step, thereby accumulating a reward of 1 at each step until termination. In particular, for all possible model-selection paths $p \in P_t$, the optimal value function satisfies

$$V^*(x, \mathbf{y}^p) = T - t.$$

Similarly, the $Q$ function

$$Q^{\pi^*}(x, y_{\leq t}, y_{t+1}) = r(x, y_{\leq t}, y_{t+1}) + T - t + 1. \tag{11}$$

In particular, for each policy $\pi_i$, define the greedy token at state $s^p$ as

$$y_i^p := \arg\max_{y \in \mathcal{Y}} \pi_i(y \mid x, \mathbf{y}^p).$$

Then, the optimal state–action value under $\pi^*$ is given by

$$Q^{\pi^*}(x, \mathbf{y}^p, y_i^p) = Q^{\pi^*}(x, \mathbf{y}^{p \oplus i}) = r(x, \mathbf{y}^p, y_i^p) + V^*(x, \mathbf{y}^{p \oplus i}) = r(x, \mathbf{y}^{p \oplus i}) + T - t + 1, \tag{12}$$

where $p \oplus i \in P_{t+1}$ denotes the extended path that follows $p$ for the first $t$ tokens and selects token $i$ at step $t + 1$.

Since we construct the reward such that $r(x, \mathbf{y}^p) = 1$ for any path $p \in P_t$ for $t \leq T/2$. Then, for any $k \leq T/2 - 1$, we know that for any $p_k \in P_k$

$$Q^{\pi^*}(x, \mathbf{y}^{p_k \oplus i}) = T - t = V^*(x, \mathbf{y}^p). \tag{13}$$

**Verfication of Assumption**    Now we verify whether three assumptions Eq. (7), Eq. (8) and Eq. (9) holds for all these MDPs.

**Eq. (7)** By the construction, we can easily know that $V^* = T$. Now, recall that $R(x, y_{\leq T}) = \sum_{i=1}^{T} r(x, y_{\leq i})$. For any token-level routing generated response (full path) $p_T$ on MDP $M_p$ ($p$ is path with length $T/2$), we have

$$V^{p_T} = R(x, \mathbf{y}^{p_T}) = \begin{cases} T - \varepsilon & \text{if } p_T \text{ has prefix } p \\ T/2 + 1 - \Delta - \varepsilon & \text{else} \end{cases}.$$

Hence, on the MDP $M_p$ with $p \in P_{T/2}$, if path $p_T \in P_T$ has prefix $p$, we know that $V^{p_T} = T - \varepsilon$, which implies that $V^* - V^{p_T} \leq \varepsilon$.

**Eq. (8)** Note that both the transition kernel and the optimal policy $\pi^*$ are deterministic. As a result, all expectations reduce to deterministic quantities. For $x, y_1, \cdots, y_T \sim \pi^*$, we know that $y_{t+1} = \arg\max_{y \in \mathcal{Y}} r(x, y_{\leq t}, y)$, which implies that

$$\mathbb{E}_{y_{t+1} \sim \pi^*(\cdot \mid x, y_{\leq t})}[Q^{\pi^*}(x, y_{\leq t+1})] = V^*(x, y_{\leq t}) = T - t. \tag{14}$$

Also, for $t \geq 1$, by Eq. (11), we have

$$\arg\max_{i \in [n]} \mathbb{E}_{y_{t+1} \sim \pi_i(\cdot \mid x, y_{\leq t})}[Q^{\pi^*}(x, y_{\leq t}, y_{t+1})] = r(x, y_{\leq t}, y_{t+1}) + T - t + 1 = T - t.$$

The last inequality is because $y_1 \in \mathcal{Y} \setminus \{y_{1,i}\}_{i \in [n]}$ and then all rewards are equal to 1. Hence, for any $t \geq 1$, we have

$$\mathbb{E}_{x, y_t \sim \pi^*} \left| \arg\max_i \mathbb{E}_{y_{t+1} \sim \pi_i(\cdot \mid x, y_{\leq t})}[Q^{\pi^*}(x, y_{\leq t+1})] - \mathbb{E}_{y_{t+1} \sim \pi^*(\cdot \mid x, y_{\leq t})}[Q^{\pi^*}(x, y_{\leq t+1})] \right| = 0 \leq \Delta.$$

Moreover, for $t = 0$, we have

$$\arg\max_{i \in [n]} \mathbb{E}_{y_1 \sim \pi_i(\cdot \mid x)}[Q^{\pi^*}(x, y_1)] = r(x, y_1) + T - 1 = (1 - \varepsilon) + T - 1 = T - \varepsilon.$$

Hence, for $t = 0$, we also have

$$\mathbb{E}_{x, y_t \sim \pi^*} \left| \arg \max_i \mathbb{E}_{y_{t+1} \sim \pi_i(\cdot | x, y_{\leq t})} \left[ Q^{\pi^*}(x, y_{\leq t+1}) \right] - \mathbb{E}_{y_{t+1} \sim \pi^*(\cdot | x, y_{\leq t})} \left[ Q^{\pi^*}(x, y_{\leq t+1}) \right] \right| = \varepsilon \leq \Delta.$$

**Eq. (9)** First, by our construction, for any prefix $x, y_{\leq t}$, one can follow the optimal policy $\pi^*$ to generate a response $x, y_{\leq T}$ such that

$$R(x, y_{\leq T}) = \sum_{i=1}^{n} r(x, y_{\leq i}) = \sum_{i=1}^{t} r(x, y_{\leq i}) + T - t. \tag{15}$$

This equality holds because optimal policy will select the reward-maximizing token at every remaining time step, that leads to a $T - t$ reward.

Now, on a particular MDP $M_p$ with path $p \in P_{T/2}$, we know that for any prefix $x, y_{\leq t}$, if there is not a full response $x, y_{\leq T}$ with prefix $x, y_{\leq t}$ and $R(x, y_{\leq T}) \geq V^{\pi^*} - \Delta = T - \Delta$, we must have

$$\sum_{i=1}^{t} r(x, y_{\leq i}) + T - t \leq T - \Delta,$$

which implies that

$$\sum_{i=1}^{t} r(x, y_{\leq i}) \leq t - \Delta. \tag{16}$$

By our construction, Eq. (16) holds if and only if $t \geq T/2 + 1$ and $y_{\leq T/2} = \mathbf{y}^p$.

Hence, we only need to verify Eq. (9) holds when $y_{\leq T/2} \neq \mathbf{y}^p$. We denote $\pi_i(x, y_{\leq t})$ is the deterministic next token given prefix $x, y_{\leq t}$ for expert model $\pi_i$. Then, we know that

$$\left| \arg \max_i \mathbb{E}_{y_{t+1} \sim \pi_i(\cdot | x, y_{\leq t})} \left[ Q^{\pi^*}(x, y_{\leq t+1}) \right] - \mathbb{E}_{y_{t+1} \sim \pi^*(\cdot | x, y_{\leq t})} \left[ Q^{\pi^*}(x, y_{\leq t+1}) \right] \right|$$

$$= \left| \arg \max_{i \in [n]} r(x, y_{\leq t}, \pi_i(x, y_{\leq t})) - 1 \right|.$$

**Situation 1:** If $t = T/2$ and $y_{\leq t} = \mathbf{y}^p$ coincides with $\mathbf{y}^p$, we have

$$\left| \arg \max_{i \in [n]} r(x, y_{\leq t}, \pi_i(x, y_{\leq t})) - 1 \right| = |(1 - \Delta) - 1| = \Delta.$$

**Situation 2:** If $t \geq 1$ and $y_{\leq T/2} \neq \mathbf{y}^p$, by our construction, we have

$$\left| \arg \max_{i \in [n]} r(x, y_{\leq t}, \pi_i(x, y_{\leq t})) - 1 \right| = 0.$$

**Situation 3:** For $t = 0$, we can have

$$\left| \arg \max_{i \in [n]} r(x, y_{\leq t}, \pi_i(x, y_{\leq t})) - 1 \right| = |(1 - \varepsilon) - 1| = \varepsilon.$$

Combining these three conditions above, we successfully verify the Eq. (9).

Since the learner only observes the expert value $Q^{\pi^*}(x, y_{\leq t})$ at the visited states $(x, y_{\leq t})$, and since these values are identical across all candidate MDPs $\{M_p\}_{p \in \mathcal{P}_{T/2}}$ for all $t \leq T/2 - 1$, the learner receives exactly the same observation sequence $o_{T/2}$ under every $M_p$. Consequently, the MDPs are indistinguishable to the learner during the first $T/2 - 1$ steps.

We consider token-level routing algorithms $\mathcal{A} : \mathcal{O} \to [n]$ whose observation space at step $t$ is defined as

$$\mathcal{O} = \bigcup_{t=1}^{T} \mathcal{O}_t = \bigcup_{t=1}^{T} \Big\{ x, \, y_{\leq t}, \, \{Q^{\pi^*}(x, y_{\leq k})\}_{k=1}^{t}, \, \{Q^{\pi^*}(x, y_{\leq t}, y)\}_{y \in \mathcal{Y}} \Big\}.$$

That is, at each step $t$, the learner's observation $o_t \in \mathcal{O}_t$ includes the prompt, the generated prefix, the expert value along the realized trajectory, and the expert values of all possible next-token continuations. Formally, let $\mathcal{A} : \mathcal{O} \to [n]$ be any (possibly randomized) token-level routing algorithm. Then there exists a path $p^* = (p_1^*, \ldots, p_{T/2}^*) \in \mathcal{P}_{T/2}$ such that

$$\mathbb{P}\big[\mathcal{A}(o_i) = p_{i+1}^* \text{ for all } i = 0, \ldots, T/2 - 1\big] \leq \frac{1}{|\mathcal{P}_{T/2}|} = \frac{1}{n^{T/2}}.$$

This follows from a simple counting argument. Since the observation sequence is identical across all $M_p$, the algorithm $\mathcal{A}$ induces a probability distribution over the set of paths $\mathcal{P}_{T/2}$, and thus

$$\sum_{p \in \mathcal{P}_{T/2}} \mathbb{P}[\mathcal{A}(o_i) = p_{i+1} \text{ for all } i = 0, \ldots, T/2 - 1] \leq 1.$$

Therefore, there must exist at least one path $p^* \in \mathcal{P}_{T/2}$ whose probability mass is at most $1/|\mathcal{P}_{T/2}|$.

Hence, for the MDP $M_{p^*}$, this expect value of this token-level routing algorithm $\mathcal{A}$ will be at most

$$(T/2 + 1 - \Delta - \varepsilon) \cdot \left(1 - \frac{1}{n^{T/2}}\right) + (T - \varepsilon) \cdot \frac{1}{n^{T/2}} \leq T/2 + 2,$$

which implies that $V^{\pi_{\mathcal{A}}} \leq V^* - T/2 + 2$. Hence, no token-level routing algorithm with observation space

$$o_t = \Big\{ x, \, y_{\leq t}, \, \{Q^{\pi^*}(x, y_{\leq k})\}_{k=1}^{t}, \, \{Q^{\pi^*}(x, y_{\leq t}, y)\}_{y \in \mathcal{Y}} \Big\}$$

can guarantee achieving an approximately optimal value for a given prompt.

$\square$

## B.2. Proof of Eq. (6) in Section 4.4

In fact, since we have

$$\mathbb{E}_{x \sim \rho, y_{\leq t} \sim \pi^*}\big[\arg\min_i \mathrm{TV}(\pi_i'(\cdot \mid x, y_{\leq t}), \pi^*(\cdot \mid x, y_{\leq t}))\big] \leq \Delta,$$

from Performance Difference Lemma 4.1, we can get

$$\mathbb{E}_{x \sim \rho}[V^{\pi^*}(x) - V^{\pi}(x)] = \sum_{t=0}^{T-1} \mathbb{E}_{x \sim \rho, y_t \sim \mathbb{P}_t^{\pi^*}(\cdot|x)}\Big[\mathbb{E}_{y_{t+1} \sim \pi(\cdot|x, y_{\leq t})} Q_h^{\pi}(x, y_{\leq t+1}) - \mathbb{E}_{y_{t+1} \sim \pi^*(\cdot|x, y_{\leq t})} Q_h^{\pi}(x, y_{\leq t+1})\Big]$$
$$\leq T \cdot Q_{\max} \cdot \mathrm{TV}(\pi \| \pi^*).$$

Since for any token, the policy $\pi$ chooses the expert $\pi_i$ such that

$$i = \arg\min_i \mathrm{TV}(\pi_i'(\cdot \mid x, y_{\leq t}), \pi^*(\cdot \mid x, y_{\leq t})) \leq \Delta,$$

we know that $\mathrm{TV}(\pi \| \pi^*) \leq \Delta$. Hence, we finally have

$$\mathbb{E}_{x \sim \rho}[V^{\pi^*}(x) - V^{\pi}(x)] \leq T Q_{\max} \mathrm{TV}(\pi \| \pi^*) \leq T Q_{\max} \Delta.$$

## C. Theoretical Discussion of Prior Token-Level Approaches

In §4, we show that training the router on SFT dataset which follows trajectories generated by $\pi^*$ can be viewed as approximating an optimal expert selection strategy. By the performance difference lemma, we establish a direct connection between the generated responses and the optimal response.

However, in recent years, Collab (Chakraborty et al., 2025) provides a test-time controlled-decoding based multi-LLM token-level collaboration approaches. During the decoding process, for each token position, Collab first generates multiple candidate tokens from multiple experts $\pi_1, \cdots, \pi_n$. For candidate tokens for each expert $\pi_i$, Collab then generates a few more tokens using $\pi_i$ for evaluating the $Q^{\pi_i}$ function. Finally, they choose the token with the maximum corresponding $Q$ function from the candidate token pool. In simple words, Collab tends to choose the action that

$$a_h = \arg\max_{a_h} \mathbb{E}_{y_{t+1} \sim \pi_{a_h}(\cdot|x,y_{\leq t})}[Q_h^{\pi_{a_h}}(x, y_{\leq t+1})].$$

Since the action is not selected by maximizing the *optimal* Q function, it leads to a mismatch

$$\min_{i \in [n]} \Delta_i(x, y_{\leq t}) = Q^*(x, y_{\leq t}) - Q^{\pi_i}(x, y_{\leq t})$$

between the generated response and the optimal response. This mismatch can lead to non-ideal performance. In fact, consider two different expert policies $\pi_1$ and $\pi_2$ with a horizon $H$ and $\pi_1(x,y) \neq \pi_2(x,y)$ for any prefix $(x,y)$, we can construct the reward function that satisfies

$$r(x, y_{\leq t}, y_{t+1}) = \mathbb{I}\{y_{t+1} = \pi_1(x, y_{\leq t})\}, \quad \forall t \leq \frac{H}{3},$$

and

$$r(x, y_{\leq t}, y_{t+1}) = \mathbb{I}\{y_{t+1} = \pi_2(x, y_{\leq t})\}, \quad \forall t > \frac{H}{3}.$$

In this setting, the optimal policy is $\pi^* = \pi_1$ for $t \leq H/2$ and $\pi^* = \pi_2$ for $t \geq H/2 + 1$, which has $Q^*(x) = H$ for prompt $x$. However, at position 0, we have

$$Q^*(x) = H, \quad Q^{\pi_1}(x) = \frac{H}{3}, \quad Q^{\pi_2}(x) = \frac{2H}{3}.$$

Hence, the mismatch error

$$\max_{i \in [n]} \Delta_i(x, y_{\leq t}) = \frac{H}{3},$$

which is linear in the horizon $H$.

Compared to Collab, FUSIONROUTE directly finetunes the router on the expert dataset, which is equivalent to

$$a_h = \arg\max_{a_h} \mathbb{E}_{y_{t+1} \sim \pi_{a_h}(\cdot|x,y_{\leq t})}[Q_h^*(x, y_{\leq t+1})],$$

which helps us to bound the final error using Performance Difference Lemma.

## D. Ablation Study

We provide the ablation study results in Section 6.1.

| Method | GSM8K | MATH500 | MBPP | HumanEval | IFEval | Avg Acc. |
|---|---|---|---|---|---|---|
| *Llama-3 Family* | | | | | | |
| Collab | 0.82 | 0.32 | 0.28 | 0.54 | 0.55 | 0.502 |
| FUSIONROUTE w/o complementary logits | 0.82 | 0.32 | 0.26 | 0.56 | 0.65 | 0.522 |
| FUSIONROUTE | **0.82** | **0.33** | **0.36** | **0.63** | **0.69** | **0.566** |
| *Gemma-2 Family* | | | | | | |
| Collab | 0.52 | 0.22 | **0.26** | 0.35 | 0.45 | 0.360 |
| FUSIONROUTE w/o complementary logits | 0.62 | **0.23** | 0.23 | 0.33 | 0.51 | 0.384 |
| FUSIONROUTE | **0.65** | 0.22 | 0.25 | **0.40** | **0.61** | **0.426** |

*Table 2.* Cross-domain performance comparison between purely token-level routing and FUSIONROUTE across model families. Best and second-best performance within each family are highlighted by **bold** and underline, respectively.

# E. Experiment Details

## E.1. Generation

### E.1.1. MIXED-DOMAIN GENERATION

Due to the large number of prompts in certain tasks and computational constraints, for each task we evaluate on a subset of their datasets. The generation settings and number of samples are summarized in Table 3.

| Dataset | # of samples | # Max New Tokens |
|---|---|---|
| GSM8K | 400 | 400 |
| MATH500 | 500 | 400 |
| MBPP | 500 | 400 |
| HumanEval | 164 | 400 |
| Ifeval | 400 | 400 |

*Table 3.* Number of samples and number of max new tokens for each dataset.

### E.1.2. GPT EVALUATION PROMPT

For GPT-4o evaluation on the general dataset, we use the following text as the prompt provided to the GPT-4o model. The output scores for both responses are then extracted to compute the win rate.

---

**GPT-4o Evaluation Prompt**

**Prompt:** You are a precise assistant for checking the quality of the answer. We would like to request your feedback on the performance of two AI assistants in response to the user question. Please rate the correctness, fluency, and level of detail of their responses. Your evaluation should consider factors such as the accuracy, relevance, depth, creativity, fluency, and level of detail of the response. Bear in mind that the response might be truncated at the end due to length constraints. Each assistant receives an overall float score on a scale of 1 to 10, in increments of 0.25. A higher score indicates better overall performance. Do not give two responses a same score. Do not prefer longer response. Please first output a single line containing only two values indicating the scores for Assistant 1 and 2, respectively. The two scores are separated by a space. In the subsequent line, please provide a short explanation of your evaluation, avoiding any potential bias and ensuring that the order in which the responses were presented does not affect your judgment.
[Question]
question
[The Start of Assistant 1's Answer]
answer1
[The End of Assistant 1's Answer]
[The Start of Assistant 2's Answer]
answer2
[The End of Assistant 2's Answer]

---

## E.2. Router's Behavior

In this section, we examine the routing behavior of our model on both domain-specific and general datasets to demonstrate the effectiveness of the router's token-level routing capability. In the following text, we use red, blue and softgreen to represent the math, coding and instruction-following model, respectively.

### E.2.1. EXAMPLE OF CODING PROBLEM

**Prompt:** Below is an instruction that describes a task. Write a response that appropriately completes the request.
Instruction: Write a function to find the minimum cost path to reach (m,n) from (0,0) for the given cost matrxi cost[][]
and a position (m,n) in cost[][]. Assistant:
**Response:** Here is a Python solution using dynamic programming:

```python
"python def minCostPath(cost, m, n):
    dp=[[0 for x in range(n)] for y in range(m)]     # Initialize the first row and first column
    for i in range(m):
        dp[i][0] = cost[i][0]
    for j in range(n):
        dp[0][j] = cost[0][j]    # Fill dp[][] in bottom up manner
    for i in range(1,m):
        for j in range(1,n):
            dp[i][j] = min(dp[i-1][j] + dp[i][j-1] + cost[i][j]
 return dp[m-1][n-1]"
```

This function works by creating a 2D array dp[][] where dp[i][j] represents the minimum cost to reach the cell (i,j)
from (0,0). The first row and first column are initialized with the cost of the cells in the first row and first column
respectively. Then, for each cell (i,j) in the rest of the matrix, the minimum cost to reach it is the minimum cost to
reach the cell above it (i-1,j) or to the left of it (i,j-1) plus the cost of the cell itself. The function finally returns the
minimum cost to reach the cell (m,n).

### E.2.2. EXAMPLE OF MATH PROBLEM

**Prompt:** Below is an instruction that describes a task. Write a response that appropriately completes the request.
Instruction: Convert the point $(0, 3)$ in rectangular coordinates to polar coordinates. Enter your answer in the form
$(r, \theta)$, where $r > 0$ and $0 \leq \theta < 2\pi$. Assistant:
**Response:** The polar coordinates $(r,\theta)$ of a point $(x,y)$ are given by $r =\sqrt {x^2+y^2}$ and
$\theta= \tan^{-1}\left(\frac{y}{x}\right)$. In this case, we have $x =0$ and $y=3$, so $r=\sqrt{0^2 + 3^2}=3$ $\theta=\tan^{-1}\left(\frac{3}{0}\right)=\frac{\pi}{2}$. Therefore, the polar coordinates of the point
$(0,3)$ are $\boxed{(3,\frac{}{2})}$. The answer is: $(3,\frac{\pi}{2})$

### E.2.3. EXAMPLE OF INSTRUCTION FOLLOWING PROBLEM

**Prompt:** Below is an instruction that describes a task. Write a response that appropriately completes the request. Instruction: Write a resume for a fresh high school graduate who is seeking their first job. Make sure to include at least 12 placeholders represented by square brackets, such as [address], [name]. Assistant:

**Response:** Here is a sample resume for a fresh high school graduate: [Name][Address][City, State Zip][Phone Number][Email Address] Objective: To secure an entry-level position in a dynamic and growth-oriented organization where I can apply my skills and knowledge to contribute to the company's success. Education: - High School Diploma, [Name of High School], [Graduation Date] - GPA: [GPA] Skills: - Proficient in Microsoft Office (Word], Excel, PowerPoint, Outlook) - Strong communication and teamwork skills - Proficient in [Programming Language or Software] - Experience with [Software or Technology] Work Experience: - [Job Title], [Company Name], [Employment Dates]  -[Briefly describe job responsibilities and accomplishments] - [Job Title], [Company Name], [Employment Dates]   - [Briefly describe job responsibilities and accomplishments] Volunteer Experience: - [Volunteer Position], [Organization], [Volunteer Dates]   - [Briefly describe volunteer work and accomplishments] References: - Available upon request. Note: This is a basic template, and you should customize it to fit your specific experiences and qualifications. Remember to proofread your resume multiple times for any errors or typos. Good luck with your job search! Please let me know if you need any further assistance. Also, please note that this is a very basic template, and you may want to consider adding more sections or details to make your resume more comprehensive. Some additional sections you may want to consider include: Extracurricular activities - Awards or recognition - Relevant coursework - Projects or presentations - Leadership roles - Relevant certifications or licenses Let me know if you need any help with any of these sections. Also, please keep in mind that this is a vaery basic template, and you may want to consider tailoring your resume to the specific job you are applying for. You can do this by highlighting...

### E.2.4. EXAMPLE OF GENERAL PROBLEM IN PERFECTBLEND DATASET

**Prompt:** Below is an instruction that describes a task. Write a response that appropriately completes the request. Instruction: You start out with some pens. Mike gives you another 22 pens. Cindy, whose father makes pens, decides to double your pens. You give Sharon 19 pens. At the end, you have 65 pens. How many pens did you start with?

**Response:** Let's denote the number of pens you started with as x. According to the problem, Mike gives you 22 pens, so you now have x + 22 pens. Cindy doubles your pens, so you now have 2(x + 22) pens. You give Sharon 19 pens, so you now have 2(x + 22) - 19 pens. According to the problem, at the end, you have 65 pens, so we can set up the equation: 2(x + 22) - 19 = 65. Now, let's solve for x: 2x + 44 - 19 = 65   2x + 25 = 65   2x=65 - 25  2x = 40  x = 40/2  x = 20  So, you started with 20 pens. The answer is: 20.

### E.2.5. DISCUSSION ON THE QUALITATIVE EXAMPLES

Based on the above qualitative examples, we observe that FUSIONROUTE successfully assigns the majority of tokens associated with specific problem types such as coding, mathematical reasoning, and instruction following, to their corresponding expert models. This behavior demonstrates the effectiveness of FUSIONROUTE in performing accurate token-level routing on domain-specific datasets, enabling the model to effectively leverage the complementary expertise of different specialists.

Furthermore, when evaluated on general-purpose datasets, we find that FUSIONROUTE exhibits a more nuanced routing pattern. Tokens that require mathematical or logical reasoning are predominantly routed to the math and code experts, while tokens related to discourse structure, formatting, or general instruction phrasing (e.g., "According to", "Let's", "So") are more frequently assigned to the instruction-following model. This adaptive token-level routing behavior highlights FUSIONROUTE's ability to dynamically balance specialized reasoning and general linguistic fluency, thereby improving overall generation quality across diverse tasks.

