# OpenReview forum: "Token-Level LLM Collaboration via FusionRoute"
_ICML.cc/2026/Conference — ICML 2026 regular_

### Official Review · Reviewer_jGzX · 2026-02-28

**Soundness:** 2
**Presentation:** 3
**Significance:** 2
**Originality:** 3
**Overall Recommendation:** 3
**Confidence:** 4

**Summary:**

This work proposes FusionRoute, which could fuse multiple LLM experts at the token level to generate better responses. Specifically, the proposed model adopts an LM as the router, and jointly trains it via both LM and router losses via SFT. Besides, the authors also conduct a CDPO stage, which enables the LM router to refine LLM expert predictions when needed. In experiments, FusionRoute has been verified based on Llama-3 and Gemma-2 on math, code, and instruction following domains.

**Compliance With Llm Reviewing Policy:**

Affirmed.

**Final Justification:**

Thanks for the rebuttal with additional experiments. The authors have addressed some of my questions.

However, I still have the following concerns:

A more advanced base model/expert is necessary because it involves the fundamental logic of FusionRoute’s effectiveness - fusing LLMs may significantly improve the performance of diverse tasks compared to a single LLM (considering the efficiency loss FusionRoute may bring).
The impact of stronger real-world post-training of experts on the above conclusions requires a more comprehensive discussion.

**Key Questions For Authors:**

-	The authors are encouraged to more comprehensively explain the necessity of token-level LLM fusion compared to one powerful LLM with better post-training. For example, could this framework achieve better results beyond the current LLM boundary? Moreover, lots of methods have been proposed to address the seesaw effect issue, such as the effort in alleviating SFT forgetting, which could be discussed.

-	The details (e.g., model size) of the LM router and experts should be given in the main content.

-	Some related references are missing [1].

[1] Ding, Ning, et al. "Fusing Highly Specialized Language Models for Comprehensive Expertise." Proceedings of the 63rd Annual Meeting of the Association for Computational Linguistics (Volume 1: Long Papers). 2025.

**Limitations:**

Yes.

**Strengths And Weaknesses:**

Strengths:

-	The idea of adopting an LM-based router to jointly 1) select better LLM experts to generate the next token, and 2) refine the output of experts is novel and sound.

-	The authors have conducted theoretical discussions in Appendix.

-	The proposed framework achieves the best overall performance compared to baselines.

Weaknesses:

-	The author merely adopts the proposed framework on Llama-3 and Gemma-2 (proposed in 2024), while there are lots of better-performing and up-to-date opensource LLMs (e.g., Qwen3) also widely used in both academy and industry. Will these LLMs also have the same seesaw effect on diverse domains? Verifying the effectiveness of FusionRoute on these evolved LLMs is essential.

-	The baseline models and the FusionRoute seem to use different trained datasets. For example, do all baselines adopt the same SFT and DPO datasets (which are informative since the Fine-tuned Model achieves very good performance among baselines)? Do all baselines benefit from the Math/Code/Instruct experts from MergeBench that may have additional domain-specific training data? The improvement of the proposed model may mainly derive from additional data.

-	The efficiency of this work should be clearly discussed. It seems that the LM-based router brings in additional cost. Will MoE models with similar inference cost achieve better results?

-	In Table 1, why do the majority of baselines achieve even worse results compared to some Experts? Fine-tuned Model achieves the overall best performance among baselines. Is it based on the original base model or the corresponding experts?

---

> ### Author Rebuttal · Authors · 2026-03-31
>
> Thanks for your insightful questions!  We will address your question as follows.
>
> > 1. Why not use other model families like Qwen3?
>
> We chose Llama-3 and Gemma-2 because MergeBench [1] provides domain-specific pretrained models for these families. For Qwen3, a practical limitation is the lack of a suitable small coding expert for our setup. We believe our method is not tied to a specific model family, so similar improvements should also transfer to other families.
>
> > 2. Why do baselines underperform the experts, and is the fine-tuned model initialized from the base model or one of the experts?
>
> The reason is that cross-domain collaboration is challenging because naively combining experts often introduces interference instead of preserving their strengths, so gains in one domain may hurt another. The Fine-tuned Model baseline, initialized from the instruction-following expert, retains decent general ability but cannot absorb other experts' strengths. This highlights FusionRoute’s key advantage: it can select the most suitable expert at each token and achieve stronger cross-domain performance.
>
> > 3. The improvement may derive from additional data.
>
> Most prior multi-agent collaboration baselines in our comparison are training-free and therefore do not use SFT or DPO data. Our contribution is to achieve strong cross-domain performance with efficient routing with a little additional training. We also show that the gains are not merely from extra training by including the Fine-tuned Model baseline.
>
> To further show that the gains are not simply from extra training data, we add the fine-tuned models to the expert pool and rerun the baselines within the Llama-3 family. Due to time limits, we report results on three benchmarks: GSM8K, HumanEval, and IFEval.
>
> | Method (add finetuned model) | GSM8K | HumanEval | IFEval | Avg. |
> |---|---:|---:|---:|---:|
> | Seq Selection | 0.79 | 0.46 | 0.61 | 0.620 |
> | TaskArithmetic | 0.82 | 0.31 | 0.38 | 0.503 |
> | DARE | 0.78 | 0.20 | 0.32 | 0.433 |
> | Collab | 0.83 | 0.49 | 0.57 | 0.63 |
> | FusionRoute | 0.82 | 0.63 | 0.69 | 0.713 |
>
> We find that FusionRoute outperforms all baselines, even when we add the fine-tuned models to the expert pools. Hence, the improvement of FusionRoute is not only from the additional data.
>
> > 4. Efficiency
> We measure the decoding speed of FusionRoute in the Llama-3 family, against both Collab and single-LLM decoding.
>
> | Method | Decoding speed (tokens/s) |
> |---|---:|
> | Collab | 1.72 |
> | FusionRoute | 11.90 |
> | Single-LLM decoding | 37.38 |
>
> These results show that FusionRoute is nearly 7× faster than Collab while getting about 3× the latency of standard single-model decoding, which is reasonable given the extra routing step and per-token cross-GPU transfer.
>
> > 5.  Will MoE achieve better results?
>
> In principle, an MoE with similar inference cost could also perform well, but it typically requires more design, end-to-end training, and usually experts with the same architecture. In contrast, FusionRoute reuses existing experts, supports different sizes and architectures, and only trains a router. As a result, it offers strong performance with lower training costs and greater flexibility.
>
>
> > 6. Could FusionRoute go beyond the LLM boundary?
>
> We believe FusionRoute can outperform strong fine-tuned models of the same size, since a single model often struggles to cover diverse domains well. We measure the performance of three popular well-finetuned variants of Llama-3.1-8B-Instruct: Llama-3.1-Tulu-3-8B, Dolphin3.0-Llama3.1-8B and NousResearch/Hermes-3-Llama-3.1-8B.
>
> | Method | GSM8K | MATH500 | MBPP | HumanEval | IFEval | Avg |
> |---|---:|---:|---:|---:|---:|---:|
> | Tulu | 0.82 | 0.27 | 0.24 | 0.56 | 0.81 | 0.540 |
> | Dolphin | 0.74 | 0.24 | 0.28 | 0.50 | 0.77 | 0.506 |
> | Hermes | 0.62 | 0.24 | 0.35 | 0.52 | 0.55 | 0.456 |
> | FusionRoute | 0.82 | 0.33 | 0.36 | 0.63 | 0.69 | 0.566 |
>
> We find that Tulu and Dolphin are strongest on IFEval, while Hermes is stronger on coding, whereas FusionRoute performs better overall across domains. FusionRoute’s lower IFEval score likely because of the weaker chosen instruction expert (0.67). Replacing it with a stronger one such as Tulu would likely improve the IFEval score.
>
>
> > 7. Training details and Missing references [2].
>
> We will add details to the main text if we have more pages in the revised version. [2] also studies cross-domain model fusion, but it mixes experts through added gating layers and end-to-end training, which requires all experts to share the same architecture. In contrast, FusionRoute supports experts with different architectures and sizes. We will add this reference in the revised version.
>
>
> Thanks again for your time and effort! We are happy to address your further concerns if you have more.
>
>
>
> [1]. He et al. MergeBench: A Benchmark for Merging Domain-Specialized LLMs
>
> [2]. Ding, Ning, et al. Fusing Highly Specialized Language Models for Comprehensive Expertise.

---

> > ### Author Rebuttal · Reviewer_jGzX · 2026-04-02
> >
> > Thanks for the rebuttal with additional experiments. The authors have addressed some of my questions.
> >
> > However, I still have the following concerns:
> > 1) A more advanced base model/expert is necessary because it involves the fundamental logic of FusionRoute’s effectiveness - fusing LLMs may significantly improve the performance of diverse tasks compared to a single LLM (considering the efficiency loss FusionRoute may bring).
> > 2) The impact of stronger real-world post-training of experts on the above conclusions requires a more comprehensive discussion.

---

> > > ### Author Response · Authors · 2026-04-04
> > >
> > > Thank you for the follow-up! These are important questions about the scope and scalability of FusionRoute's value proposition. We address both points below.
> > >
> > > For the first point (on stronger base models/experts), we would like to clarify that the main goal of FusionRoute is not to outperform the best possible single LLM that could be obtained with sufficiently large-scale data and post-training. Rather, our goal is to build a training-friendly and efficient collaborative system that can leverage already available domain-specialized expert models and naturally absorb their complementary strengths. In this sense, FusionRoute is designed for the practical regime where one has access to strong experts from different domains, and the key question is how to utilize them efficiently, rather than how to train a single model from scratch.
> > >
> > > We also believe that our experiments on both the Llama-3 and Gemma-2 families already validate the effectiveness of FusionRoute in combining expert knowledge, and the final performance in Llama-3 family outperforms existing popular variants of LLM fine-tuned from Llama-3.1-8B-Instruct. Hence, we believe the same advantage remain under stronger base models as well.
> > >
> > > For the second point (on stronger real-world post-training of experts), if future post-training yields a single model that uniformly dominates all state-of-the-art experts at comparable size and inference cost, then the room for multi-expert collaboration would naturally shrink. However, we believe this regime is difficult to realize in practice, especially when the experts are also the same size. It would be hard for a single model to simultaneously absorb the strengths of multiple strongly post-trained domain experts while avoiding cross-domain interference.

---

### Official Review · Reviewer_bhYm · 2026-03-11

**Soundness:** 3
**Presentation:** 2
**Significance:** 3
**Originality:** 3
**Overall Recommendation:** 3
**Confidence:** 4

**Summary:**

The paper presents FusionRoute, a method for multi-LLM collaboration at the token-level based on a mixture of experts approach.  A theoretical analysis is presented showing that expert-only routing is limited unless strong global coverage assumptions hold.  This motivates augmenting expert selection with a trainable complementary generator.

**Compliance With Llm Reviewing Policy:**

Affirmed.

**Final Justification:**

This is an interesting direction, but I still have some concerns that were raised in the original review that are not fully addressed in the author response.

**Key Questions For Authors:**

- What is the high-level intuition for why the complementary logits should be expected to improve performance over a traditional MoE router?
- How does the proposed method compare to the simple ensembling approach mentioned above?  That approach seems to have an advantage that it is much simpler and doesn't require training a router.

**Limitations:**

Yes

**Strengths And Weaknesses:**

Strengths:
- The paper presents an alternative way to combine models complementary to model merging and agents.
- The idea of using a complementary logits in a MoE model appears to be novel.

Weaknesses
- A large portion of the paper is based on a theoretical analysis of MoE models at the token level.  While each part of this analysis seems to make sense, the paper is lacking an intuitive explanation for why adding the complementary logits are a good idea.  It's also a bit unclear to me whether this theoretical analysis is really needed for this type of paper - it seems preferable to just have a clear intuitive explanation of why this approach should be expected to work better, combined with empirical evidence to support the claim that the proposed method outperforms the current state of the art.
- The empirical analysis is a bit on the lighter side.  Also more details could be included - e.g., how is the best response selected in sequence selection.
- The paper could compare to stronger baselines, for instance ensembling the models' predictions.  E.g., combine the logits form each model token by token without using a router.  This will require more computation of course, but I do think it is important to compare to.  It might also help to report some measure of the compute required for each method.

---

> ### Author Rebuttal · Authors · 2026-03-31
>
> Thanks for your detailed review and insightful questions!  We will address your question as follows.
>
> > 1. Intuitive explanation for why adding the complementary logits is a good idea.
>
> The intuition behind the complementary logits is that they help prevent truly poor token choices during decoding. For example, for some prefixes, none of the candidate experts may offer a sufficiently good next-token prediction, or the router may be not accurate and choose a wrong model. In such cases, the router can supply complementary logits to correct or refine the expert distribution, steering decoding toward a reasonably good token instead of being limited by the experts alone and choosing a very bad token. We will add this intuition in our revised version.
>
>
> > 2. The empirical analysis is a bit on the lighter side.
>
> We add several experiments to make the evaluation more comprehensive:
> (a) a mix-logits baseline and efficiency comparisons across methods, to better position FusionRoute against other token-level approaches;
> (b) state-of-the-art reward models for response selection in the sequence-selection baseline;
> (c) 500 additional prompts from OpenHermes to further verify the robustness of FusionRoute’s generation quality; (point 4 in rebuttal for jNo9)
> (d) comparisons with a fine-tuned baseline, showing that the gains do not come solely from additional training data. (point 2 in rebuttal for jGzX)
>
> Combining with our experiments and ablation studies in the paper,  we think our experiment is comprehensive enough to show the effectiveness of FusionRoute.
>
> > 3. How is the best response selected in sequence selection.
>
> We feed responses to the external reward model to get a score for each model's response, and choose the best one. In our main paper, we choose the same reward function as [1] (Ray2333/reward-model-Mistral-7B-instruct-Unified-Feedback).  We additionally use the current best reward model in RewardBench2 (Skywork/Skywork-Reward-V2-Llama-3.1-8B). and the results are shown below.
>
> | Method | GSM8K | MATH500 | MBPP | HumanEval | IFEval | Avg. |
> |---|---:|---:|---:|---:|---:|---:|
> | Sequence Selection (New) | 0.75 | 0.33 | 0.33 | 0.48 | 0.51 | 0.480 |
> | FusionRoute | 0.82 | 0.33 | 0.36 | 0.63 | 0.69 | 0.566 |
>
> We find that sequence selection still performs poorly even when using a very strong reward model, which further highlights the effectiveness of FusionRoute.
>
>
>
> > 4. The paper could compare to stronger baselines such as combine the logits token by token without using a router.
>
> Thanks for raising this point! We try the mix logits approach for the Llama-3 family.
>
> | Method | GSM8K | MATH | MBPP | HumanEval | IFEval | Avg. |
> |---|---:|---:|---:|---:|---:|---:|
> | Mix logits | 0.82 | 0.31 | 0.34 | 0.57 | 0.59 | 0.526 |
> | FusionRoute | 0.82 | 0.33 | 0.36 | 0.63 | 0.69 | 0.566 |
>
> We find that mix-logits is a strong baseline, but it still performs clearly worse than FusionRoute. It also requires a forward pass through every expert at each token, leading to higher computational overhead.
>
>
> > 5. measure of computation.
>
> We compare the inference speed of FusionRoute in the Llama-3 family, against both the prior token-level collaboration method Collab and standard single-LLM decoding. We run 10 prompts and measure the average number of tokens generated per second. The measured generation speeds are:
>
> | Method | Decoding speed (tokens/s) |
> |---|---:|
> | Collab | 1.72 |
> | Mix Logit | 8.51 |
> | FusionRoute | 11.90 |
> | Single-LLM decoding | 37.38 |
>
> These results show that FusionRoute is nearly 7× faster than Collab while staying within about 3× the latency of standard single-model decoding, which we view as reasonable given the extra routing step and per-token cross-GPU transfer. There is also room to further reduce this overhead.
>
> The mix-logits approach is also slower than FusionRoute, and its decoding cost grows roughly linearly with the number of experts because it combines all experts’ logits at every step. In contrast, FusionRoute’s decoding speed is largely independent of the number of experts, since it routes to only one selected expert at inference time.
>
> Thanks again for your time and effort! We are happy to solve your further concerns if you have more.
>
>
>
> [1]. Chakraborty et al. Collab: Controlled Decoding using Mixture of Agents for LLM Alignment

---

> > ### Author Rebuttal · Reviewer_bhYm · 2026-04-01
> >
> > Thanks for the response.  This addresses some of my concerns, but I still am not totally clear on what is the intuition for what is the intuition for why adding complementary logits would be expected to work better, and since this seems to be the main contribution in this paper, I would expect a bit more of a clear story (probably with some analysis) on that.  I think this is probably hard to address in the author response and probably needs a revision of the paper to make clear.

---

> > > ### Author Response · Authors · 2026-04-04
> > >
> > > Thanks for the follow-up! We would love to provide a more clear analysis here. The motivation comes from the observation that existing token-level collaboration mechniasms may fail many cases, which we believe complementary logits will help a lot.
> > >
> > > First, the router can be inaccurate at some particular token, and the coressponding experts fail to generate the correct token. Here is an example. Consider a math prompt whose next token should be 42 after the prefix “What is 6 * 7? The answer is”, and the logits of expert and router are shown in the following Table.
> > >
> > > | Candidate next token | Math expert | Code expert | Instruct expert (selected) | Router complementary | Final = selected expert + router |
> > > |---|---:|---:|---:|---:|---:|
> > > | `42` | 3.6 | 3 | 2.0 | 2.0 | **4.0** |
> > > | `21` | 1.5 | 1.0 | 1 | 2.5 | 3.5 |
> > > | `derived` | -1.0 | 0.1 | **2.6** | -0.2 | 2.4 |
> > >
> > > Here, the selected expert alone would have produced 'derived', which is wrong. However, after logit addition, the final distribution shifts to 42. Intuitively, in this case, the router can help to **correct the distribution by adding the complementary logits**.
> > >
> > > Second, there might be some prefixes that none of them is good for generating the next token. For example, consider a prompt whose next token should be $B$ after the prefix
> > > > "What is 17 + 25?
> > > A. 40
> > > B. 41
> > > C. 42
> > > D. 43
> > > Answer:"
> > >
> > > The possible logits of expert and router are shown in the following Table.
> > >
> > > | Candidate next token | Math expert (Selected) | Code expert | Instruct expert | Router complementary | Final = selected expert + router |
> > > |---|---:|---:|---:|---:|---:|
> > > | `B` | 2.5 | 2 | 1 | 2.5 | **5.0** |
> > > | `42` | **3** | -0.5 | -0.5 | 1 | 4 |
> > > | `C` | -1 | 3.5 | 3 | 3 | 2 |
> > >
> > > We can see that even if the router is still not correct for generating the next token, its complementary logits can help the final system to generate the correct token.
> > >
> > >
> > > Hence, both cases illustrate why complementary logits are useful: **they help when the router makes an inaccurate expert selection, and they also help when none of the experts is sufficient on its own**. We also provide an ablation study below (more details in Appendix E), showing that removing the complementary logits consistently leads to worse multi-domain performance than FusionRoute.
> > >
> > > | Variant | Avg. Acc. | $\Delta$ vs. Fine-tuned |
> > > |---|---|---|
> > > | Fine-tuned model (same SFT+DPO, no routing) | 53.6% | -- |
> > > | FusionRoute w/o complementary logits (routing only) | 52.2% | -1.4% |
> > > | FusionRoute (routing + complementary logits) | 56.6% | +3% |

---

### Official Review · Reviewer_pzr2 · 2026-03-11

**Soundness:** 2
**Presentation:** 3
**Significance:** 3
**Originality:** 3
**Overall Recommendation:** 4
**Confidence:** 3

**Summary:**

This paper proposes FusionRoute, a token-level collaboration framework for multiple domain-specialized LLMs. At each decoding step, a trained router selects the most suitable expert model and adds its own complementary logits to refine the expert’s next-token prediction. The paper combines this routing-plus-correction framework with a theoretical analysis of the limitations of pure token-level expert selection, and evaluates the method on Llama-3 and Gemma-2 expert families across math, coding, and instruction-following tasks. The results show improvements over sequence-level collaboration, prior token-level collaboration, model merging, and direct fine-tuning baselines.

**Compliance With Llm Reviewing Policy:**

Affirmed.

**Final Justification:**

ok

**Key Questions For Authors:**

1.Does FusionRoute require all experts to come from the same model family and share the same tokenizer and vocabulary? Would the method still work for heterogeneous experts across different families?

2.The router outputs both routing weights and complementary logits. Could the authors clarify why these two roles can be learned stably by the same router, and how mixed training prevents interference between them?

3.Why do the experiments only cover Llama-3 and Gemma-2, but not other major open-source model families such as Qwen? Is this mainly due to resource constraints, or does the method rely on specific compatibility conditions?

4.The paper emphasizes efficiency over prior multi-model collaboration methods. Could the authors provide more direct efficiency comparisons, such as inference latency, throughput, or additional computational overhead?

**Limitations:**

No. The discussion on limitations and social impacts was too brief. This article should more explicitly address the assumptions regarding expert compatibility, generalization of heterogeneous models, additional inference costs, limited experimental coverage, and the risks of increased system complexity and difficulty in explaining failure modes in multi-LLM deployments.

**Strengths And Weaknesses:**

From the soundness perspective, the paper presents a reasonably complete method, and the theoretical analysis, main experiments, and ablations are broadly consistent with the main claims. That said, there is still some gap between the theoretical assumptions and realistic LLM settings, and the experiments are limited to specific model families and expert combinations. In terms of presentation, the paper is generally well structured and easy to follow, although the differences from prior token-level collaboration methods could be stated more directly. In terms of significance, the paper studies a practical and relevant problem, namely how to efficiently combine multiple LLMs with different expertise, although the utility of the approach depends on the availability of strong expert models. In terms of originality, combining expert routing with complementary logits in a single router is an interesting idea, and the theoretical discussion of the limitations of pure routing is useful; however, the method overall appears more like a strong integration of existing ideas than a fundamentally new paradigm.

---

> ### Author Rebuttal · Authors · 2026-03-31
>
> Thanks for your positive review and insightful questions!  We will address your question as follows.
>
> > 1. Does FusionRoute require all experts to come from the same model family and share the same tokenizer and vocabulary? Would the method still work for heterogeneous experts across different families?
>
> Yes. The current FusionRoute requires the model with the same tokenizer and vocabulary. This is also an intrinsic limitation for many token-level approaches such as logit distillation [1], speculative decoding [2], and previous token-level collaboration approaches [3]. However, FusionRoute supports expert models with different sizes and architectures, rather than requiring experts with the same size as in model merging, Mixture-of-Expert system and some model fusion approaches [4].
>
> > 2. The router outputs both routing weights and complementary logits. Could the authors clarify why these two roles can be learned stably by the same router, and how mixed training prevents interference between them?
>
> Thank you for raising this point. Both routing weight and complementary logits share the same backbone because both require the same contextual understanding ability, so it is possible to be learned stably by the same router backbone but different output layers.
>
> The mix training is designed to solve the interference problem: During the CDPO phase, if you only use DPO loss to train the router backbone for generating complementary logits, the linear layer of the routing weight will remain unchanged and become inconsistent with the router backbone. It will eventually influence the routing's ability. Hence, we use mix training to **maintain the routing ability when training the router's complement logits**, and it finally prevents this interference problem.
>
>
> > 3. Why do the experiments only cover Llama-3 and Gemma-2, but not other major open-source model families such as Qwen? Is this mainly due to resource constraints, or does the method rely on specific compatibility conditions?
>
> We chose Llama-3 and Gemma-2 because MergeBench [5] already provides strong domain-specific pretrained models for these families. For Qwen, one practical reason is that it is not easy to find a good small coder model (the smallest well-known coding-specialized model is 30B), which is not suitable in our experiments. More broadly, our method does not rely on any special property of Llama-3 or Gemma-2; we selected them because they are both major open-source model families and are well supported by existing merged-model study.
>
> > 4. The paper emphasizes efficiency over prior multi-model collaboration methods. Could the authors provide more direct efficiency comparisons, such as inference latency, throughput, or additional computational overhead?
>
> We compare the inference speed of FusionRoute in the Llama-3 family, against both the prior token-level collaboration method Collab and standard single-LLM decoding. We run 10 prompts and measure the average number of tokens generated per second. The measured generation speeds are:
>
> | Method | Decoding speed (tokens/s) |
> |---|---:|
> | Collab | 1.72 |
> | FusionRoute | 11.90 |
> | Single-LLM decoding | 37.38 |
>
> These results show that FusionRoute is nearly 7× faster than Collab, providing clear quantitative evidence of a substantial efficiency improvement over prior token-level collaboration approaches. At the same time, FusionRoute remains within roughly 3× the latency of standard single-model decoding, which we believe is reasonable given the additional routing step and per-token cross-GPU transfer overhead. We believe there is still room for further improvement, potentially reducing the overhead to around 2×.
>
> > 5. Limitations
>
> Thanks for raising this point. We will add all limitations proposed above, such as the same tokenizer requirement, additional inference cost and the complexity of the system in our revised version.
>
> Thanks again for your time and effort! We are happy to solve further concerns if you have more.
>
>
> [1]. Agarwal et al. On-Policy Distillation of Language Models: Learning from Self-Generated Mistakes
>
> [2]. Leviathan et al. Fast Inference from Transformers via Speculative Decoding.
>
> [3]. Shen et al. Learning to Decode Collaboratively with Multiple Language Models
>
> [4]. Ding et al. Fusing Highly Specialized Language Models for Comprehensive Expertise.
>
> [5]. He et al. MergeBench: A Benchmark for Merging Domain-Specialized LLMs

---

> > ### Author Rebuttal · Reviewer_pzr2 · 2026-04-01
> >
> > Thank you for your reply. I will keep my score unchanged.

---

> > > ### Author Response · Authors · 2026-04-04
> > >
> > > Thank you for your time and effort! We greatly appreciate your insightful questions and suggestions, which have helped us improve the paper.

---

### Official Review · Reviewer_jNo9 · 2026-03-21

**Soundness:** 3
**Presentation:** 3
**Significance:** 3
**Originality:** 3
**Overall Recommendation:** 4
**Confidence:** 4

**Summary:**

This paper focuses on how to combine expert models into a general-purpose system. The authors argue that existing technical routes are all suboptimal: MoE requires additional training and architectural compatibility; model merging may introduce parameter interference; and multi-agent collaboration leads to low inference efficiency. Building on the collaboration paradigm, the paper proposes dynamically selecting which expert model should take over at the token level, while also allowing the system to correct expert errors on its own.
The proposed method, FUSIONROUTE, centers on a lightweight router. Given a prefix, the router outputs routing weights over expert models together with a set of compensation logits. During inference, the next-token probability distribution is obtained by combining the output of the expert models selected by the router with the compensation logits. Training is conducted in two stages, SFT and DPO, with the expert models frozen in the second stage. In addition, the paper provides a theoretical analysis from the perspective of MDP, showing that the compensation logits can help reduce the error bound of overall system performance.

**Compliance With Llm Reviewing Policy:**

Affirmed.

**Final Justification:**

I think most of my concerns have been addressed, so I am increasing my score from 3 to 4.

**Key Questions For Authors:**

- I believe the paper is missing a crucial ablation. The router does not function solely as a selector; it also compensates as part of the expert output. This makes it unclear whether the observed gains come from the collaboration mechanism itself or simply from adding an extra generator trained with SFT and DPO.

**Limitations:**

yes

**Strengths And Weaknesses:**

**Strengths:**

* The paper is clearly written, and the motivation is well articulated.
* FUSIONROUTE appears to show strong advantages on larger models.
* The idea of compensation logits is sensible and well motivated.

**Weaknesses:**

* The paper describes the method as being applicable to “heterogeneous, off-the-shelf LLMs,” which feels somewhat overstated. The experiments only validate the approach on experts from the same model family, without considering a truly heterogeneous expert pool with clear differences in architecture or output style.
* The paper presents efficiency as one of its core contributions, but does not provide sufficiently strong quantitative evidence to support this claim. Moreover, the method is essentially an extension of an MoE-style system to token-level routing, so it is not obvious that it should bring additional efficiency gains.
* The theoretical analysis reads more like a post hoc interpretation, and its assumptions are rather strong. In particular, the setup is highly idealized: it assumes fixed-length trajectories, deterministic policies, deterministic transitions, treats the training data as if it were generated by an optimal policy, and further introduces a fairly strong global coverage assumption.
* In the general evaluation, the paper samples 500 prompts from the PerfectBlend test set. This is not a large sample, and the paper does not report the corresponding random seed or variance. As a result, it is difficult to rule out the possibility that the results are partly due to randomness.

---

> ### Author Rebuttal · Authors · 2026-03-31
>
> Thanks for your detailed review and insightful questions!  We will address your question as follows.
>
> > 1.  Only validate on experts from the same model family
>
> We acknowledge that FusionRoute requires all experts to share the same tokenizer and vocabulary. This is a common limitation of token-level methods, including speculative decoding and prior token-level collaboration approaches [3]. Still, FusionRoute allows experts with different sizes and architectures, unlike model merging and many MoE methods. Moreover, even with a shared tokenizer, experts can still differ substantially in behavior when trained with different objectives.
>
> > 2. Effiency
>
> We compare the inference speed of FusionRoute against Collab, and standard single-LLM decoding. The measured generation speeds are:
>
> | Method | Decoding speed (tokens/s) |
> |---|---:|
> | Collab | 1.72 |
> | FusionRoute | 11.90 |
> | Single-LLM decoding | 37.38 |
>
> These results show that FusionRoute is nearly 7× faster than Collab while staying within about 3× the latency of standard single-model decoding, which we view as reasonable given the extra routing step and per-token cross-GPU transfer. There is also room to further reduce this overhead.
>
> Moreover, unlike MoE, which typically requires end-to-end training of the whole system, FusionRoute only trains the router and reuses existing experts. This makes it cheaper to train and more flexible, especially since it supports experts with different architectures.
>
> > 3. The setup of theoretical analysis is highly idealized: fixed-length trajectories, deterministic policies and transitions, training data distribution assumption, and a global coverage assumption.
>
> We will address your theoretical concerns one-by-one.
>
> (a). Fixed-length trajectory: This assumption is mainly for notational simplicity and is not fundamental. In the token-level MDP, a state whose prefix already contains EOS can be treated as terminal, so variable-length generation is naturally supported. Thus, the fixed-horizon setup is only a convenient presentation.
>
> (b). Deterministic policy and transition: This can be relaxed to stochastic policies without materially changing the analysis in Sections 4.1, 4.2, and 4.4. In Theorem 4.3, determinism is used only to construct the counterexample in the lower-bound proof, so it is better viewed as a proof device rather than a core assumption. Also, in LLM decoding, the transition is always deterministic: $\mathbb{P}(s_{t+1}\mid s_t, y_{t+1}) = 1$ only for $s_{t+1}=(x,y_{\le t+1})$ and otherwise 0.
>
> (d). Treats the training data as if it were generated by an optimal policy:
> We do not need to assume the training data comes from an optimal policy. It can instead be generated by some policy $\pi^{\text{data}}$, in which case the gap terms $V^{\pi^*}-V^\pi$ in the theory become $V^{\pi^{\text{data}}}-V^\pi$. Then the guarantee is no longer near-optimality, but closeness to the performance of $\pi^{\text{data}}$.
>
> (e). Global coverage assumption: We agree that this is a strong assumption. As stated in Section 4.2, it requires that for every prefix, at least one expert can provide a sufficiently good next token, which is not universally realistic. Our analysis uses this assumption as a starting point rather than an endpoint: Section 4.3 shows that lacking such coverage is an inherent limitation of prior token-level collaboration methods, while Section 4.4 shows that router-logit complementation can relax this requirement. This helps explain why the additional router signal is necessary.
>
>
> > 4. More prompts for general evaluation
>
> We additionally evaluate on 500 prompts from the OpenHermes dataset. The win-rate results are shown below. We observe a similar trend in this additional evaluation set.
>
> | Algorithm | Win rate (vs. fine-tuned model) |
> |---|---:|
> | Collab | 42.6% |
> | Model Selection | 49.0% |
> | FusionRoute | 58.1% |
>
> We also argue that evaluating on 500- 1000 prompts is not considered small. For example, popular benchmarks such as AlpacaEval, WildBench, Arena-Hard-Auto, and previous work [2] all use roughly 500--1000 prompts. Hence, we believe this comparison reasonably represents the quality differences among the algorithms.
>
> > 5. Whether the observed gains come from adding an extra generator trained with SFT and DPO.
>
> Our “Fine-tuned model” baseline is already trained with SFT and DPO on the same dataset. We include it precisely to test whether the gains come only from additional training. Since its win rate and multi-domain performance are both worse than FusionRoute, the results suggest that the gains come not just from training, but also from the collaboration mechanism and router-logit complementation.
>
> Thanks again for your time and efforts! We will be happy to answer any further questions you may have.
>
> [1]. Shen et al. Learning to Decode Collaboratively with Multiple Language Models
>
> [2]. Chakraborty et al. Collab: Controlled Decoding using Mixture of Agents for LLM Alignment

---

> > ### Author Rebuttal · Reviewer_jNo9 · 2026-04-03
> >
> > Thank you for the rebuttal. Regarding W5, the clarification shows that the improvements cannot be simply attributed to additional training.
> > However, from a more fine-grained attribution perspective, I believe it is still not fully clear how much of the gain comes from the collaboration mechanism itself, and how much comes from the additional expressive power introduced by router-logit complementation / the final distribution. Providing a more detailed analysis would be helpful.
> >
> > Overall, I think most of my concerns have been addressed, so I am increasing my score from 3 to 4.

---

> > > ### Author Response · Authors · 2026-04-04
> > >
> > > Thank you for raising your score! We provide a more fine-grained quantitative attribution below.
> > >
> > > Using the Llama-3 family as an example, we can decompose the gains additively using three data points from our ablations:
> > >
> > > | Variant | Avg. Acc. | $\Delta$ vs. Fine-tuned |
> > > |---|---|---|
> > > | Fine-tuned model (same SFT+DPO, no routing) | 53.6% | -- |
> > > | FusionRoute w/o complementary logits (routing only) | 52.2% | -1.4% |
> > > | FusionRoute (routing + complementary logits) | 56.6% | +3% |
> > >
> > > This decomposition reveals three important findings:
> > >
> > > 1. **Token-level routing alone does not surpass fine-tuning.**
> > > The routing-only variant (52.2%) slightly underperforms the fine-tuned model (53.6%). Note that this is not a weakness of our paper—it is precisely the empirical manifestation of the impossibility result in Theorem 4.3: pure expert selection, even when trained, cannot reliably recover a near-optimal policy because it is limited to the fixed action space of the experts. We believe this is the first empirical confirmation of this theoretical limitation.
> > >
> > > 2. **The complementary logits provide the decisive gain.**
> > > Adding the router's complementary logits to the routing-only variant yields a 4.4% improvement (52.2% → 56.6%), which accounts for the entirety of the net gain over the fine-tuned baseline. The same pattern holds for Gemma-2 (38.4% → 42.6%, a +4.2% gain from complementary logits). This confirms the theoretical benefit shown in Section 4.4: augmenting expert logits with a trainable complementary component expands the effective policy class and enables recovery of near-optimal value.
> > >
> > > 3. **Routing and complementation are synergistic, not independent.**
> > > Although routing alone does not beat fine-tuning, it plays an essential structural role: it selects the domain-appropriate expert whose logits the router then complements. Without routing, the complementary component would need to correct *all experts' weaknesses* simultaneously rather than correcting only the best-matched expert's residual errors. This is why FusionRoute (56.6%) substantially outperforms the fine-tuned model (53.6%), even though the fine-tuned model also benefits from the same training data and optimization—the fine-tuned model lacks the ability to dynamically select which expert to refine.
> > >
> > >
> > > Overall, these results indicate that both the collaborative mechanism and the router's complementary logits play important roles in achieving strong performance, and both contribute substantially to the final gains.

---

### Decision · Program_Chairs · 2026-04-30

**Decision:**

Accept (regular)

**Comment:**

This paper studies token-level routing. The basic setup is that there are a pool of models. A trained router selects the model to provide the next token along with a contribution coming from the router itself for when the selected model is incorrect.

Token-level routing is an important problem, and the idea of collaboration the authors bring up, along with the notion of using the router itself to additionally provide signal (through the complementary logits) is pretty interesting. Results are solid as well.

Reviewers were a little bit mixed but largely positive on this paper. In terms of drawbacks, there’s some concerns with whether as promised the paper really applies to heterogeneous models, whether there is enough overall experimental evidence for the claims, and similarly whether the authors do enough to explore what the the complementary token mechanism really provides.

This paper is definitely promising—it could use a bit more work in terms of validating and exploring its basic claims, but contains the kernel of something potentially valuable.